# An Illusion of Progress? Assessing the Current State of Web Agents

**Tianci Xue**♣  **Weijian Qi**♣,*  **Tianneng Shi**♠,*  **Chan Hee Song**♣  **Boyu Gou**♣

**Dawn Song**♠  **Huan Sun**♣,†  **Yu Su**♣,†

♣The Ohio State University
♠University of California, Berkeley
{xue.681, sun.397, su.809}@osu.edu
https://github.com/OSU-NLP-Group/Online-Mind2Web

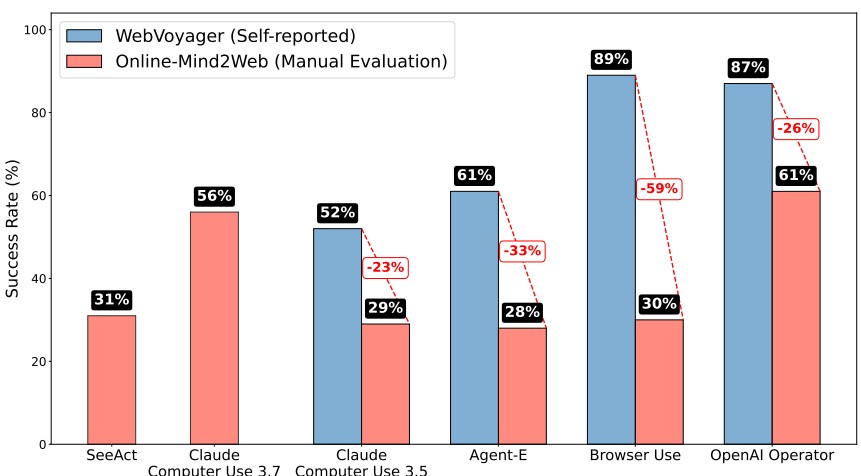

Figure 1: Frontier web agents show a drastic drop in success rate when evaluated on Online-Mind2Web (human evaluation) compared with those reported on WebVoyager (He et al., 2024a). Surprisingly, many recent agents, except for Claude Computer Use 3.7 and Operator, do not outperform the simple SeeAct agent (Zheng et al., 2024) released in early 2024.

## Abstract

As digitalization and cloud technologies evolve, the web is becoming increasingly important in the modern society. Autonomous web agents based on large language models (LLMs) hold a great potential in work automation. It is therefore important to accurately measure and monitor the progression of their capabilities. In this work, we conduct a comprehensive and rigorous assessment of the current state of web agents. Our results depict a very different picture of the competency of current agents, suggesting over-optimism in previously reported results. This gap can be attributed to shortcomings in existing benchmarks. We introduce Online-Mind2Web, an online evaluation benchmark consisting of 300 diverse and realistic tasks spanning 136 websites. It enables us to evaluate web agents under a setting that approximates how real users use these agents. To facilitate more scalable evaluation and development, we also develop a novel LLM-as-a-Judge automatic evaluation method and show that it can achieve around 85% agreement with human judgment, substantially higher than existing methods. Finally, we present the first comprehensive comparative analysis of current web agents, highlighting both their strengths and limitations to inspire future research.

---

*Equal contributions
†Corresponding authors

# 1  Introduction

Language agents that integrate large language models (LLMs) to reason and communicate via language (Su et al., 2024; Sumers et al., 2023) have quickly risen to the center stage of AI research and application. This new generation of AI agents can operate in the digital world, such as browsing the web (Deng et al., 2023; Zhou et al., 2024a; Zheng et al., 2024) or using a computer (Xie et al., 2024; Wu et al., 2024; Anthropic, 2024), like humans do. Accurately measuring and monitoring the progression of their capabilities is therefore critical because of their potential in disruptive automation and job displacement.

Recently, the field has seen several surprising results, claiming to achieve close to 90% success rate (OpenAI, 2025; Müller & Žunič, 2024) on the WebVoyager (He et al., 2024a) benchmark for web agents. That has led to great enthusiasm and optimism. However, as a scientific field, we must caution against over-optimism, especially when the supporting data may be insufficient or biased, because that leads to short-sightedness, unrealistic expectations, and irrational decisions.

In this work, we aim to conduct a rigorous assessment of the current state of web agents. A major obstacle is the benchmark. To get a comprehensive and accurate assessment of the competency of web agents, we should minimize the simulation-to-reality gap in the evaluation. It should be under a setting that approximates how real users use such agents as much as possible, which means evaluating them on realistic tasks across a wide range of real-world websites. However, most existing benchmarks either focus on offline evaluation with cached snapshots of websites (Deng et al., 2023; Lu et al., 2024) or sandbox environments with a limited number of simulated websites (Yao et al., 2022; Zhou et al., 2024a; Koh et al., 2024a). Among the benchmarks that focus on online evaluation (He et al., 2024a; Yoran et al., 2024; Pan et al., 2024b), WebVoyager is the most widely used. However, as we will discuss in detail later (§2.1), WebVoyager has several shortcomings: (1) It lacks coverage and diversity in tasks and websites, (2) many tasks have shortcut solutions such that a simple agent that primarily uses Google Search can already solve up to 51% of the tasks, and (3) its LLM-as-a-Judge automatic evaluation has a low agreement with human judgment. These issues together lead to substantially inflated evaluation results (Figure 1). Therefore, a better online benchmark for web agents is needed for our assessment.

To this end, the main contributions of this work are three-fold:

1. We introduce a new benchmark, **Online-Mind2Web**, that contains 300 diverse and realistic tasks spanning 136 websites. We conduct careful manual evaluation of six frontier web agents, and the results depict a drastically different picture about the competency of current agents (Figure 1). Many recent agents, except for Claude Computer Use 3.7 (Anthropic, 2025) and Operator (OpenAI, 2025), underperform the simple SeeAct agent (Zheng et al., 2024) released in early 2024. Even Operator only achieves a success rate of 61%, showing substantial room for improvement.

2. As human evaluation is not scalable, to facilitate future agent development and evaluation, we develop a new automatic evaluation, **WebJudge**, based on LLM-as-a-judge (Zheng et al., 2023). We show that it can reach around 85% agreement with human judgment and an average success rate gap of only 3.8%, significantly outperforming existing approaches. Agent ranking under our automatic evaluation also closely aligns with human evaluation. Moreover, WebJudge demonstrates excellent generalization capabilities, achieving high precision and a small success rate gap across five well-known out-of-domain benchmarks. It is therefore a useful tool for rapid iteration on agent development and evaluation.

3. We also conduct the first **comprehensive comparative analysis** on the current web agents, which leads to novel insights on their respective advantages and limitations and sheds light on further improvement.

## 2 New Online-Mind2Web Benchmark

### 2.1 Why Introduce a New Benchmark?

To get an accurate assessment of the competency of web agents, we need to evaluate them on realistic tasks across a wide range of real-world websites under a setting that approximates how real users use such agents as much as possible. However, existing benchmarks fail to meet these criteria in several ways:

• Some benchmarks, such as the original Mind2Web (Deng et al., 2023) or WebLINX (Lu et al., 2024), adopt an offline setting by caching portions of websites. While this setting facilitates rapid iteration during agent development, it inevitably suffers from incompleteness due to the dynamic nature of many websites. As a result, agents are restricted from exploring the environment and can only follow the annotated reference trajectory.

• Sandbox environments like WebShop (Yao et al., 2022), (Visual-)WebArena (Zhou et al., 2024a; Koh et al., 2024a) mitigate the exploration issue, but the diversity of websites is inherently limited due to the sheer difficulty of creating full replica of modern websites.

• A few benchmarks like WebVoyager (He et al., 2024a), AssistantBench (Yoran et al., 2024), and Mind2Web-Live (Pan et al., 2024b) focus on the online setting, evaluating web agents on live, real-world websites. To facilitate automatic evaluation, AssistantBench only includes time-insensitive tasks, i.e., tasks with a closed-form, time-invariant answer string. While this is a reasonable compromise, it precludes the evaluation of agents on time-sensitive tasks requiring up-to-date information or procedural tasks that do not expect an answer string. Both WebVoyager and Mind2Web-Live are derived from the original Mind2Web benchmark. Mind2Web-Live includes only around 100 tasks, and its key node-based evaluation is still prone to changes in websites over time, making it less reliable.

Given the widespread use of WebVoyager in recent agent releases (Abuelsaad et al., 2024; Müller & Žunič, 2024; Anthropic, 2024; OpenAI, 2025; Zhou et al., 2024b; He et al., 2024b; Azam et al., 2024; H Company, 2025; DeepMind, 2024), we conduct a more detailed analysis. The tasks in WebVoyager are synthesized by using modified tasks from Mind2Web as seeds and prompting a language model to generate additional ones. Moreover, its coverage is limited to just 15 websites. Upon closer inspection, we find that the tasks are generally simple—potentially due to limitations in the task synthesis pipeline—and many of them require minimal navigation or interaction with the website.

To quantify this observation, we develop a naive search agent that follows two simple steps: (1) generate a query for Google Search, and (2) click into a returned link to check the presence of the answer—without performing any further operations on the website. We randomly sample 100 tasks from WebVoyager, stratified by website, and manually evaluate the results. Surprisingly, this simple search agent can already achieve a 51% success rate. This confirms our observation that the tasks in WebVoyager are skewed toward the easier end, and that many can be solved using shortcuts such as Google Search instead of navigating the websites. These factors—including the limited diversity of tasks and websites, along with issues in automatic evaluation (to be discussed in Section 3)—help explain the substantial discrepancy observed in Figure 1.

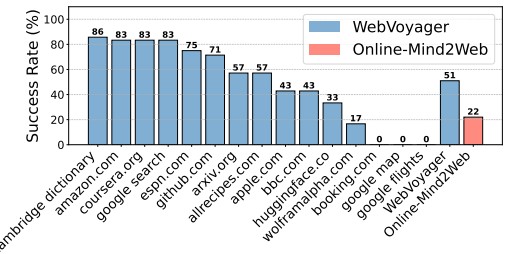

Figure 2: Success rate of the simple search agent on WebVoyager vs. Online-Mind2Web.

### 2.2 Dataset Construction

Given the shortcomings of existing benchmarks, we collect a new dataset tailored to enable rigorous and accurate evaluation of web agents. The ever-evolving nature of websites makes evaluating web agents on online tasks challenging, as there does not always exist a fixed

ground truth. To handle this issue, we first systematically verify the validity of tasks in existing datasets and construct an up-to-date dataset by filtering out: (1) **Ambiguous tasks:** tasks with unclear or vague instructions, leading to multiple possible interpretations. (2) **Invalid tasks:** tasks that are no longer executable due to structural changes in websites, removal of necessary features or outdated, resulting in the inability to obtain valid results. (3) **CAPTCHA-protected websites:** tasks involving websites with strong bot protection prevent agents from completing the task.

We begin by randomly selecting 650 tasks from the original Mind2Web dataset and evaluating their viability. Our analysis shows that 47% of the tasks are either invalid or have outdated ground-truth trajectories. Then, we construct our dataset by selecting 167 tasks from the original Mind2Web dataset that are as distinct as possible, rewriting another 24 tasks from Mind2Web, incorporating 34 tasks from Mind2Web-Live,[1] and manually creating 75 new tasks on websites with high traffic.[2] It is worth highlighting that the reality and diversity of the original Mind2Web tasks stem from their being crowdsourced and rigorously validated. Following this principle, we adopt a similar procedure to ensure our tasks remain both realistic and broadly representative. For each rewritten task, we primarily modify task requirements to guarantee solvability or refine the task description to eliminate ambiguity.

In total, we present a comprehensive benchmark of 300 high-quality and realistic tasks spanning 136 popular websites from various domains, designed to evaluate web agents on real-world settings systematically. To obtain a fine-grained assessment of agents' performance, we categorize tasks into three levels of difficulty based on the number of steps $N_{step}$ required for a human annotator to complete them, which we refer to as *reference length*. Tasks with $N_{step} \leq 5$ are labeled as easy, $6 \leq N_{step} \leq 10$ as medium, and $N_{step} \geq 11$ as hard. Finally, we have 83 easy tasks, 143 medium tasks and 74 hard tasks. More details about distribution and illustrative examples can be found in Appendix C.2 and G.1. We are committed to maintaining the benchmark over time. If any tasks become outdated or infeasible, we will replace them with new ones of similar difficulty level to ensure fair comparison across versions.

We also evaluate the search agent on our benchmark, comprising 33 easy, 34 medium, and 33 hard-level tasks. In contrast with WebVoyager, the search agent can only solve 22% of tasks on our benchmark, with success rates for easy, medium, and hard tasks nearly 50%, 18%, and 3%, respectively, highlighting the difficulty of our benchmark. Overall, our tasks are sourced from real-world users, resulting in greater diversity and realism, whereas synthetic tasks often exhibit biases and similarity (Chen et al., 2024; Yu et al., 2023). Moreover, because our dataset encompasses a broad spectrum of popular websites, it more accurately captures the distribution of real-world scenarios. This is particularly important as tasks with identical objectives can differ significantly in complexity depending on the design and structure of the target website. Therefore, our benchmark provides a more accurate and realistic evaluation of web agents' performance in real-world scenarios.

## 3   WebJudge

Evaluating web agents in online environments is essential for a realistic performance assessment. However, conducting evaluations in unconstrained online settings presents significant challenges. Human evaluations are labor-intensive and do not scale effectively, while existing automatic methods such as rule-based and LLM-as-a-judge are still unreliable. Specifically, strict rule-based success criteria are sensitive to website changes, such as functional updates or URL modifications. Meanwhile, existing LLM-as-a-judge approaches still exhibit low agreement with human judgements (see Table 3), which may stem from either considering only the final screenshot, thereby overlooking crucial intermediate steps, or processing all screenshots within the trajectory, resulting in token overload and exceeding the context size limit. To further enhance the reliability and scalability of the evaluation process, we propose a new automatic evaluation method called WebJudge that preserves

---

[1]The tasks in Mind2Web-Live (Pan et al., 2024b) are also adapted from Mind2Web and share a comparable level of difficulty.

[2]According to https://similarweb.com/.

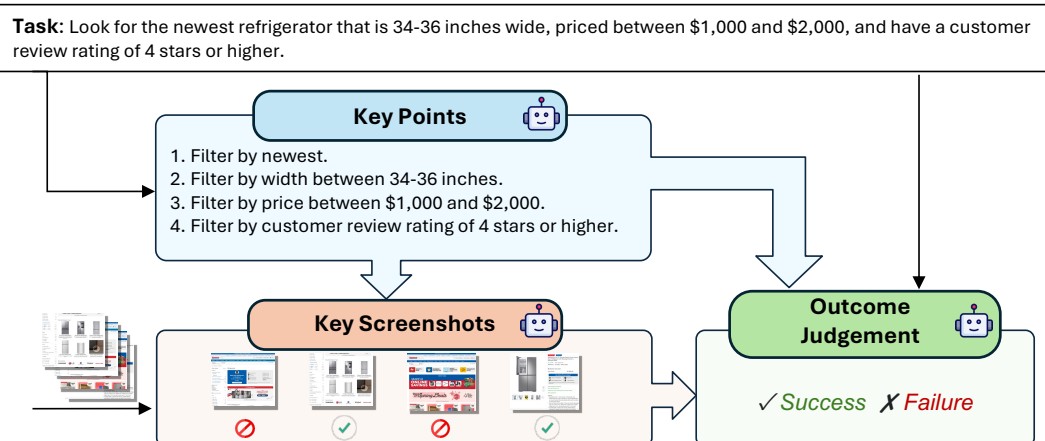

Figure 3: Illustration of **WebJudge**. (1) Key Point Identification: The model is prompted to identify several key points that are necessary for completing the task, based on the given task description. (2) Key Screenshot Identification: From a sequence of screenshots, key ones are selected to retain relevant visual evidence while discarding uninformative frames. (3) Outcome Judgement: Output the judgement result based on the task description, key points, key screenshots, and the action history.

critical intermediate screenshots while mitigating the token overload issue. Specifically, given a task description $T$, a sequence of actions $A = (a_1, a_2, \cdots, a_n)$ and a trajectory comprising a sequence of screenshots $I = (i_1, i_2, \cdots, i_n)$, the Evaluator performs a binary classification to determine the outcome $O$ as either "Success" or "Failure":

$$O = \text{Evaluator}(T, A, I). \tag{1}$$

Specifically, our LLM-as-a-judge method consists of three steps:

1. **Key Point Identification:** Typically, a task may involve several key requirements that the model must satisfy. Therefore, we first prompt the model to identify the key points $K = (k_1, k_2, ..., k_m)$ required for successful task completion based on the task description. For instance, completing a restaurant-related task may necessitate specifying the location and rating, which are considered critical key points.

2. **Key Screenshot Identification:** Then, the model generates a descriptive summary for each screenshot and evaluates their relevance ($[1, 5]$ with 5 being the highest) to task completion. Screenshots with a score above or equal to a threshold $\delta$ are filtered as key screenshots. This process allows the model to identify and focus on important intermediate steps for evaluation while not consuming too many tokens or exceeding the context size limit.

3. **Outcome Judgement:** Finally, the Evaluator integrates identified key points, selected key screenshots, task description, and the action sequence to make a comprehensive judgment of task completion. We also examine how varying levels of granularity in WebJudge's outcome judgment stage affect evaluation reliability. Specifically, comparing a chain-of-thought mode with a more fine-grained method that assesses each key point individually, where a task is deemed successful only if all key points are completed. See Appendix B.1 for details.

We compare our proposed method, WebJudge, with existing approaches, including Autonomous Evaluation (Pan et al., 2024a), AgentTrek (Xu et al., 2025), and WebVoyager (He et al., 2024a). Different evaluators require varying inputs to conduct the evaluation. Table 1 provides a detailed overview of the input requirements for each automatic evaluator. It's worth noting that not all agents provide intermediate thoughts (e.g., Operator), and some agents (e.g., SeeAct) also do not return the final response. To ensure broader applicability and fairness in evaluation, we choose not to rely on the final response. Additionally, we observe that the final response is prone to contain hallucinations, which negatively impacts evaluation (See Appendix F.3 for examples). Although it is well-known that LLMs potentially exhibit self-preference bias during evaluations (Wataoka et al., 2024; Panickssery et al.,

2024), this concern is mitigated in our setting: the LLM primarily judges environment observations (i.e., screenshots along an agent's trajectory) rather than text or images generated by an LLM. As a result, the bias issue is less pronounced.

| Evaluators | Screenshots | Action History | w/o Intermediate Thoughts | w/o Final Response |
|---|---|---|---|---|
| Autonomous Evaluation | ✓ | ✓ | ✓ | ✓ |
| AgentTrek | ✓ | ✓ | ✗ | ✓ |
| WebVoyager | ✓ | ✗ | ✓ | ✗ |
| WebJudge (Ours) | ✓ | ✓ | ✓ | ✓ |

Table 1: Input requirements for various automatic evaluation methods.

While WebJudge effectively identifies and preserves critical intermediate screenshots, its computational cost and latency may limit its scalability. Specifically, the number of API calls scales proportionally with the number of screenshots in a trajectory. As a result, for agents that generate particularly long trajectories (e.g., Operator), the evaluation process will be costly. To reduce cost and improve reliability, we also aim to train a lightweight key screenshot identification model, **WebJudge-7B**, based on Qwen2.5-VL-7B (Bai et al., 2025), which offers lower inference cost while maintaining comparable or even achieving higher agreement with human judgment. See Appendix E for details.

## 4 Experiments and Results

### 4.1 Experimental Setup

We evaluate six prominent web agents: SeeAct (Zheng et al., 2024), Browser Use (Müller & Žunič, 2024), Agent-E (Abuelsaad et al., 2024), Claude Computer Use 3.5 (Anthropic, 2024), Claude Computer Use 3.7 (Anthropic, 2025), and Operator (OpenAI, 2025). To better understand how well agents navigate on different websites and ensure fair comparisons, we initialize each task with a start URL and prompt agents not to use Google Search to prevent shortcuts. See Appendix B.2 for more detailed discussions about those constraints.

We use human annotation as the reference standard to rigorously evaluate the performance of agents and the effectiveness of automatic evaluators. We manually annotate all the trajectories of six agents based on task descriptions, action history, and screenshots. Each task is labeled by at least two annotators, with a third resolving any conflicts. We measure agreement between all automatic evaluation methods and humans on final binary decisions.

See Appendix E for implementation details of both web agents and automatic evaluators.

### 4.2 Main Results

| Agent | Human Eval | WebJudge (GPT-4o) | | WebJudge (o4-mini) | | WebJudge-7B | |
|---|---|---|---|---|---|---|---|
| | | SR | AR | SR | AR | SR | AR |
| SeeAct | 30.7 | 39.8 | 86.7 | 30.0 | 85.3 | 28.0 | 86.0 |
| Agent-E | 28.0 | 34.7 | 86.0 | 27.0 | 86.3 | 26.0 | 87.3 |
| Browser Use | 30.0 | 40.1 | 81.4 | 26.0 | 89.3 | 24.3 | 88.3 |
| Claude Computer Use 3.5 | 29.0 | 35.8 | 86.3 | 24.0 | 87.0 | 26.0 | 89.7 |
| Claude Computer Use 3.7 | 56.3 | 59.7 | 79.1 | 47.3 | 82.3 | 48.7 | 84.3 |
| OpenAI Operator | **61.3** | **71.8** | 81.8 | **58.3** | 83.7 | **59.0** | 86.3 |
| Avg Gap/AR | | 7.8 | 83.6 | 3.8 | 85.7 | 3.9 | 87.0 |

Table 2: Agent Success Rate (SR) as measured by human evaluation and WebJudge, along with Agreement Rate (AR) between WebJudge and human evaluation.

As shown in Table 2, Claude Computer Use 3.7 and Operator stand out with a success rate of 56.3% and 61%, respectively, whereas the other agents achieve a similar success rate around 30%. These results stand in stark contrast to previously reported performance on WebVoyager (See Figure 1). We believe these results are better reflective of the competency

of current web agents. The gap between WebJudge-predicted and human-evaluated success rates narrows as the underlying model's reasoning ability improves. Notably, WebJudge powered by o4-mini achieves 85.7% agreement with a success rate gap of only 3.8%, demonstrating strong alignment with human judgment. Similarly, WebJudge-7B achieves a higher agreement of 87%, also with a small gap of 3.9%, while further reducing the number of API calls to only two per trajectory. Despite setting the temperature to 0, some inherent randomness remains in the evaluator (i.e., GPT-4o). To assess the robustness of WebJudge, we also execute the evaluation pipeline 3 times. The results shown in Figure B.1 demonstrate that WebJudge exhibits high robustness with a low variance. Specifically, the average standard deviation of the six agents' success rate is 1.1%, with a maximum of 1.8%.

We also break down agent performance by task difficulty to gain a more fine-grained understanding. The results in Figure 4 show a significant drop in performance as task complexity increases. Specifically, the average success rate decreases by 31.6% when moving from easy to medium tasks, followed by an additional 15.4% drop from medium to hard tasks. Claude Computer Use 3.5 demonstrates a relatively stronger performance on hard tasks than open-source agents but lags behind on medium tasks. Claude Computer Use 3.7 and Operator achieve a high success rate of 90.4% and 83.1% on easy tasks but still struggle with harder tasks, potentially due to insufficient long-horizon planning and exploration abilities.

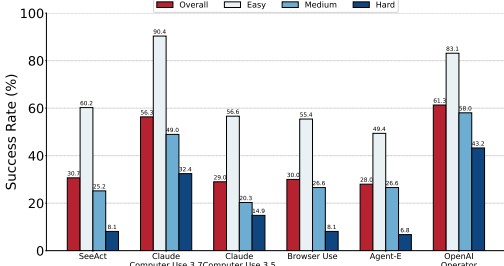

Figure 4: Agent success rate by task difficulty. All agents struggle with harder tasks.

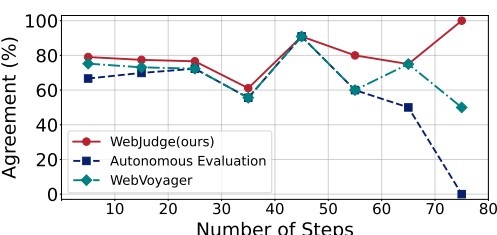

Figure 5: Agreement between WebJudge and human evaluation with respect to the number of action steps by Operator.

## 4.3 Comparison against Existing Evaluation Methods

We evaluate the agreement with human judgment for different automatic evaluation methods powered by the general-purpose GPT-4o and the reasoning-focused o4-mini. The results in Table 3 demonstrate that WebJudge consistently achieves the highest agreement with an average of 83.6% and 85.7%, respectively. WebJudge not only exhibits a minimal gap in success rates but also produces agent rankings that closely align with human evaluation (Table 2). In addition, it is generally applicable to all output formats of web agents, making it a useful tool for agent development and evaluation to complement human evaluation.

| Model | Auto-Eval | SeeAct | Agent-E | Browser Use | Claude 3.5 | Claude 3.7 | Operator | Avg AR |
|---|---|---|---|---|---|---|---|---|
| GPT-4o | Autonomous Eval | 84.7 | 85.0 | 76.0 | 83.7 | 75.5 | 71.7 | 79.4 |
| | AgentTrek Eval | 73.0 | 64.3 | 63.3 | – | – | – | 66.9 |
| | WebVoyager | – | 75.3 | 71.3 | 74.0 | 72.0 | 76.7 | 73.9 |
| | WebJudge | 86.7 | 86.0 | 81.4 | 86.3 | 79.1 | 81.8 | **83.6** |
| o4-mini | Autonomous Eval | 79.7 | 85.7 | 86.0 | 84.3 | 68.0 | 73.3 | 79.5 |
| | WebVoyager | – | 80.3 | 79.0 | 81.7 | 74.3 | 78.3 | 78.7 |
| | WebJudge | 85.3 | 86.3 | 89.3 | 87.0 | 82.3 | 83.7 | **85.7** |
| | WebJudge-7B | 86.0 | 87.3 | 88.3 | 89.7 | 84.3 | 86.3 | **87.0** |

Table 3: Agreement Rate (AR) among different automatic evaluation methods across six agents.

Notably, WebVoyager's evaluation shows a relatively low agreement rate. We find that a primary factor is the presence of hallucination in the agent's final responses (See Appendix F.3

for examples), which frequently leads to a high false positive rate during the evaluation process. This may also be a contributor to the seemingly high results previously reported on WebVoyager (e.g., Browser Use's evaluation was based on WebVoyager's automatic evaluation). Similarly, the evaluation in Pan et al. (2024a) also shows a lower agreement as it considers only the final screenshot, disregarding intermediate screenshots that are crucial for assessing task completion. This phenomenon is especially pronounced in the Operator, which tends to generate longer trajectories, often exceeding 100 screenshots.

To further illustrate these problems, we evaluate how the agreement changes with the number of action steps from Operator's trajectories. As the trajectory length increases, the human agreement of both WebVoyager and Autonomous Evaluation decline significantly. This is due to inherent limitations in their methods: WebVoyager suffers from token overload caused by an excessive number of screenshots, while Autonomous Evaluation overlooks important intermediate steps by focusing solely on the final screenshot. In contrast, our method maintains a relatively high agreement even when the number of action steps reaches 80. This result highlights the effectiveness of our key screenshot identification strategy, which reduces the number of screenshots while preserving critical intermediate information.

## 4.4 The Generalization of WebJudge

To further demonstrate the generalizability and effectiveness of WebJudge, we evaluate it on AgentRewardBench (Lù et al., 2025), which comprises 1,302 trajectories spanning 5 popular benchmarks: WebArena (WA), VisualWebArena (VWA), AssistantBench (AB), WorkArena (Work), and WorkArena++ (Wk++). As shown in Table 4, WebJudge(GPT-4o), WebJudge-7B, and WebJudge(o4-mini) significantly outperform existing methods by relying solely on screenshots and action history as input, achieving impressive overall precision of 73.7% 75.7% and 82.0%, respectively. The high precision suggests that WebJudge holds potential as a robust and scalable reward model for downstream applications such as Rejection Sampling Fine-Tuning (RFT) (Liu et al., 2024a; Yuan et al., 2023), Reflection (Shinn et al., 2023; Xue et al., 2023; Paul et al., 2024), and Reinforcement Learning (RL) (Ziegler et al., 2019; Ouyang et al., 2022), where it could serve to provide reliable feedback or filter out noisy trajectories. We leave these for future work. Moreover, the results in Table 5 suggest that success rates predicted by WebJudge(GPT-4o) align more closely with human judgments than those of existing methods, including official rule-based evaluation. Specifically, WebJudge achieves an average gap of only 5.9%, which is significantly smaller than that of the rule-based method (9.8%) and the existing approach (8.1%). While WebJudge (o4-mini) achieves exceptionally high precision comparable to the rule-based method and offers greater scalability, it also tends to be overly conservative, resulting in relatively low recall. This strictness leads to a larger success rate gap compared to human evaluation. In contrast, WebJudge-7B strikes a balance, achieving both high precision (75.7%) and a smaller success rate gap (6.0%), while significantly reducing costs to a fixed two calls per trajectory, making it a reliable and scalable evaluator for further research. Overall, these findings further emphasize the effectiveness and reliability of WebJudge as an automatic evaluator, making it a valuable tool for rapid iteration in agent development and evaluation (e.g., RFT, RL and reflection).

| Methods | AB | VWA | WA | Work | Wk++ | Overall |
|---|---|---|---|---|---|---|
| *Rule-based** | 25.0 | **85.2** | 79.0 | 100.0 | 83.3 | 83.8 |
| Autonomous Eval* | 83.3 | 61.2 | 67.6 | 96.4 | 59.3 | 67.6 |
| GPT-4o (A11y Tree)* | 77.8 | 63.0 | 70.2 | 94.6 | 63.0 | 69.8 |
| WebJudge (GPT-4o) | 66.7 | 69.8 | 72.6 | 92.3 | 75.0 | 73.7 |
| WebJudge-7B | 80.0 | 66.7 | 77.5 | 100.0 | 70.0 | 75.7 |
| WebJudge (o4-mini) | **100.0** | 74.5 | **81.2** | **100.0** | **90.0** | **82.0** |

Table 4: We follow previous settings and report the results of precision on AgentReward-Bench (Lù et al., 2025). To ensure a fair comparison , we utilize the same version of GPT-4o (gpt-4o-2024-11-20) for WebJudge. GPT-4o (A11y Tree) is the previous best result based on the accessibility tree. * indicates results are taken from Lù et al. (2025).

| Agent | Human Eval* | | | GPT-4o (A11y Tree)* | | | Rule-based* | | | WebJudge (GPT-4o) | | | WebJudge (o4-mini) | | | WebJudge-7B | | |
|---|---|---|---|---|---|---|---|---|---|---|---|---|---|---|---|---|---|---|
| | VWA | WA | Wk++ | VWA | WA | Wk++ | VWA | WA | Wk++ | VWA | WA | Wk++ | VWA | WA | Wk++ | VWA | WA | Wk++ |
| Claude 3.7 | 28.3 | 55.1 | 18.4 | 34.8 | 64.1 | 20.7 | 23.9 | 30.8 | 8.1 | 31.5 | 52.6 | 16.1 | 18.5 | 33.3 | 3.5 | 22.8 | 44.9 | 10.3 |
| GPT-4o | 35.9 | 42.3 | 18.4 | 47.8 | 50.0 | 11.5 | 17.4 | 25.6 | 4.6 | 31.5 | 46.2 | 9.2 | 20.7 | 32.1 | 3.5 | 29.4 | 38.5 | 4.6 |
| Llama 3.3 | – | 22.4 | 9.2 | – | 27.6 | 5.8 | – | 18.4 | 3.5 | – | 30.3 | 3.5 | – | 14.5 | 1.2 | – | 23.7 | 1.2 |
| Qwen2.5-VL | 21.7 | 33.3 | 13.8 | 34.8 | 52.6 | 14.9 | 17.4 | 29.5 | 11.5 | 30.4 | 44.9 | 8.1 | 20.7 | 29.5 | 3.5 | 22.8 | 35.9 | 6.9 |
| *Gap* | – | – | – | 10.5 | 10.3 | 3.4 | 9.1 | 12.2 | 8.0 | 5.4 | 6.5 | 5.7 | 8.7 | 10.9 | 12.0 | 4.4 | 4.5 | 9.2 |
| *Avg Gap* | – | – | – | 8.1 | | | 9.8 | | | **5.9** | | | 10.5 | | | 6.0 | | |

Table 5: Comparison of web agent success rates evaluated by humans, GPT-4o (A11y Tree), rule-based and WebJudge across various benchmarks. * indicates results are taken from Lù et al. (2025).

# 5 Analysis

## 5.1 Agent Efficiency

Since different web agents may require different numbers of steps to complete the same task, we introduce an efficiency metric $E$. Specifically, let $S_i$ be the number of steps an agent takes for task $i$, and $\hat{S}i$ be the corresponding reference length. The efficiency score $E$ is defined as the average ratio across the success set $\mathcal{T}_{\text{succ}}$. A lower value of $E$ indicates greater efficiency:

$$E = \frac{1}{|\mathcal{T}_{\text{succ}}|} \sum_{i \in \mathcal{T}_{\text{succ}}} \frac{S_i}{\hat{S}_i}. \tag{2}$$

Figure 6 shows the efficiency score of each agent, as well as the average number of action steps taken on successful and failed tasks. We identify two key findings: 1) **Longer agent trajectories at failed tasks:** Agents take notably more steps for failed tasks, primarily due to repeated actions or unexpected pop-up windows. In Browser Use, Claude Computer Use 3.7 and Operator, failed tasks involve nearly twice as many steps as successful ones. 2) **Exploration vs. exploitation trade-off:** Operator heavily favors exploration, i.e., extensively exploring a website on the fly, to maximize task completion while trading off efficiency ($2.6\times$ human reference, up to 44 minutes for challenging tasks). Other agents tend to directly commit to a certain (greedy) strategy, achieving a higher efficiency ($1.0\times$ human) but prone to getting stuck at dead ends.

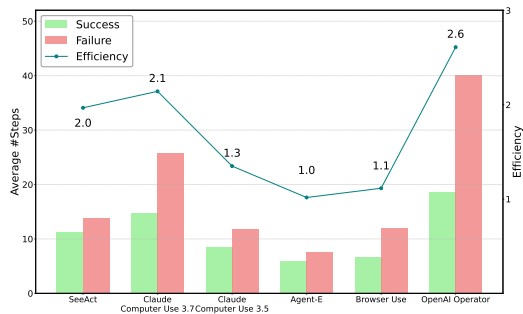

Figure 6: Average number of steps and efficiency score across agents. For efficiency score, the lower, the better.

Figure 7: Operator's error distribution. Operator mainly struggles with Filter & Sorting (57.7%) and Navigation (19.6%).

## 5.2 Error Analysis and More Discussion

To gain a fine-grained understanding of agents' limitations, we manually analyze the error cases and categorize them into the following types:

- **Filter & Sorting Errors**: Applying incorrect filters or sorting options, or failing to apply them when required.

- **Incomplete Steps**: Omitting critical steps, e.g., not clicking the "Submit" button after filling out a form, or failing to open detailed pages after a search.

- **Navigation Errors**: Deviating from the intended navigation sequence.

- **Misunderstanding**: Failing to grasp the main goal of the task entirely.

- **Others**: Rare or uncategorized failure cases.

We focus our detailed analysis on *Operator*, as it represents the state-of-the-art performance among existing web agents (see Figure 7). We first highlight key advantages that contribute to Operator's strong overall performance, followed by an analysis of its primary limitations as well as those of other agents.

**Limitations of Operator.** Although Operator has certain advantages, it also exhibits two notable limitations. First, it frequently fails to satisfy numerical and temporal constraints specified in the task instructions, either by overlooking or applying incorrect value ranges. This observation aligns with prior work indicating that LLMs are sensitive to numerical inputs (Jain et al., 2023; Qian et al., 2023). Second, despite its generally exploratory behavior, Operator sometimes misses niche website features required to complete certain tasks. See Appendix F.1.4 for examples.

**Limitations of Other Agents.** In contrast to Operator, other agents exhibit a different set of failure modes. These agents frequently neglect task requirements and often hallucinate unmet constraints. They also display limited exploration ability and perform repetitive behavior, such as prematurely terminating or redundantly repeating actions when confronted with uncertainty or interruptions. Furthermore, they tend to rely excessively on keyword-based search strategies, which leads to suboptimal results. Appendix F.2 provides further instances of such failures.

## 6 Conclusions

We introduce Online-Mind2Web, a comprehensive and realistic benchmark designed to rigorously assess the performance of web agents. Through extensive human evaluations, we find that existing frontier agents still struggle with online tasks, as most agents successfully complete only 30% of them. To enable scalable and reliable evaluation, we further propose a novel automatic evaluation method that identifies and preserves critical intermediate screenshots while mitigating the token overload issue. Our approach achieves the highest agreement with human judgments among existing methods and demonstrates superior precision, making it a promising reward model for downstream applications such as RFT, RL, and reflection. Finally, we present an in-depth analysis and highlight several key limitations of these agents, including sensitivity to numerical or temporal constraints, lack of exploration ability, and over-reliance on keyword-based search.

## 7 Acknowledgments

The authors would thank colleagues from the OSU NLP group for constructive feedback. This research was sponsored in part by NSF CAREER #1942980, the Berkeley Center for Responsible Decentralized Intelligence (RDI), and gifts from Cisco, Orby AI, and Amazon. We also appreciate computational resources provided by the Ohio Supercomputer Center. The views and conclusions contained herein are those of the authors and should not be interpreted as representing the official policies, either expressed or implied, of the U.S. government. The U.S. Government is authorized to reproduce and distribute reprints for Government purposes notwithstanding any copyright notice herein.

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

# Table of Contents in Appendix

## A   Related Work

### A.1   Web Agents and Benchmarks

Autonomous web agents have rapidly evolved from simple simulated settings (Shi et al., 2017; Humphreys et al., 2022) to real-world applications (Yao et al., 2022; Deng et al., 2023; Zhou et al., 2024a). Numerous studies have aimed to enhance agent capabilities (Hong et al., 2024; Gur et al., 2024; Zheng et al., 2024; Koh et al., 2024b; Gou et al., 2025; Gu et al., 2024; Furuta et al., 2024; Lai et al., 2024; Liu et al., 2024b; Qi et al., 2025). Despite these advances, existing benchmarks (Lu et al., 2024; Yoran et al., 2024; Pan et al., 2024b) fall short of the key desiderata for robust evaluation—namely being challenging, realistic, diverse, and reliable. Evaluation efforts remain predominantly centered on Mind2Web (Deng et al., 2023) and (Visual-)WebArena (Zhou et al., 2024a; Koh et al., 2024a), representing the most widely used offline and sandboxed online environments, respectively. Meanwhile, growing commercial interest in web agents (Abuelsaad et al., 2024; Müller & Žunič, 2024; OpenAI, 2025) has brought increased attention to WebVoyager (He et al., 2024a) due to its evaluation on online websites. However, the high success rates (~90%) reported by recent agents raise concerns about the difficulty and reliability of the benchmark. Motivated by the need for a more rigorous assessment of recent agents, we introduce Online-Mind2Web, a realistic online benchmark paired with a novel automatic evaluation framework, designed to enable accurate and scalable assessments aligned with real-world performance.

### A.2   Automatic Evaluation for Web Agents

Unlike offline settings (Deng et al., 2023; Lu et al., 2024), online evaluation is inherently challenging. SeeAct (Zheng et al., 2024) conducts first human evaluation of Mind2Web tasks on live websites. In addition, several automatic evaluation methods have been proposed, based either on rule-based heuristics (Zhou et al., 2024a; Pan et al., 2024b) or LLM-as-a-judge approaches (Zheng et al., 2023; Li et al., 2023; Fernandes et al., 2023; Bai et al., 2023). Specifically, Pan et al. (2024a) employ an MLLM to evaluate task completion via prompting. However, it only considers the final screenshot, ignoring intermediate states and leading to significant information loss. WebVoyager (He et al., 2024a) includes all screenshots, but suffers from token overload. AgentTrek (Xu et al., 2025) leverages GPT-4o to filter low-quality trajectories based on task descriptions, actions, and reasoning traces, yet our empirical analysis shows that its agreement with human judgment remains low. Rule-based methods such as Mind2Web-Live (Pan et al., 2024b) define key nodes (e.g., specific URLs or elements) per task, but are limited by annotation quality, sensitivity to webpage updates, small scale, and poor scalability. AssistantBench (Yoran et al., 2024) focuses on information-seeking tasks with static answers and evaluates performance using F1 overlap with gold-standard answer tokens, limiting its applicability to open-ended web interactions. To overcome these limitations, we propose WebJudge, a novel LLM-based evaluation framework that improves upon prior LLM-as-a-judge methods, enabling flexible and more reliable evaluation of web agents in online settings.

## B   Experimental Details

### B.1   The impact of evaluation granularity: WebJudge (CoT) vs. WebJudge (Keypoints-wise)

We also explore how different levels of granularity in WebJudge's outcome judgment stage affect evaluation reliability. There are two different levels of granularity of WebJudge: **WebJudge (CoT)** generates a final binary outcome (i.e., success or failure) using chain-of-thought reasoning based on the task description, key points, and key screenshots. In contrast, the more fine-grained **WebJudge (Keypoint-wise)** mode evaluates each key point individually in the outcome judgment phase, and a task is only considered successful if all key points are completed. As shown in the Table B.1, when the number of key points is 3 or less(the average number of key points is 3.6), WebJudge (Keypoint-wise) achieves a higher agreement rate of 87.9%. However, when the number of key points exceeds 3,

the WebJudge(CoT) performs better. This is because the generated key points are not always accurate or necessary. Evaluating each key point individually can lead to overly strict assessments, thereby lowering the agreement rate. Therefore, enabling the agent to dynamically adjust key points while interacting with the website may yield more reliable evaluations, as it gains a deeper understanding of the task and the website structure. We leave it as future work. We also found that combining both modes can further improve the overall agreement rate to 84.6%. WebJudge (Combined) leverages the strengths of each mode while mitigating their respective weaknesses, resulting in a more reliable evaluation.

We also compare the cost of the two modes of WebJudge. Since WebJudge (Keypoint-wise) evaluates each key point individually, its cost during the outcome judgment phase scales proportionally with the number of key points. In contrast, WebJudge (CoT) requires only a single call, resulting in approximately half the total cost of WebJudge (Keypoint-wise). Considering the trade-off between performance and cost, we use WebJudge (CoT) for all experiments in the paper.

| Task | WebJudge (CoT) | WebJudge (Keypoint-wise) | WebJudge (Combined) |
|---|---|---|---|
| Key Points $\leq$ 3 | 86.1 | **87.9** | - |
| Key Points > 3 | **81.4** | 76.3 | - |
| Overall | 83.7 | 81.7 | **84.6** |

Table B.1: The agreement rates with humans across different modes of WebJudge.

| Auto-Eval | Key Point Identification | Key Screenshot Identification | Outcome Judgment | Total Tokens |
|---|---|---|---|---|
| WebJudge (CoT) | 241 | 47k | 20k | **67k** |
| WebJudge (Keypoint-wise) | 241 | 47k | 73k | 120k |

Table B.2: Comparison of token costs per task across different WebJudge modes.

## B.2 The impact of Google Search constraint

There are two main reasons why we restrict agents from using Google Search.

• First, we aim to evaluate the agent's navigation capabilities as much as possible while minimizing the influence of other factors, such as shortcutting via Google search.

• Second, and more importantly, we observe that without this restriction, agents tend to use Google Search to navigate to other alternative websites when they encounter difficulties on the specified one, resulting in task completion on entirely different websites. This will lead to unfair comparisons among agents, as different websites have varying designs and functions, which may impact the difficulty of the task. To eliminate such confounding factors and enable a fair, apples-to-apples comparison, we prohibit the use of Google Search.

However, using search is sometimes a natural strategy for solving web tasks. To better understand its impact, we also evaluate the Browser-Use agent (the one that is most likely to take shortcuts via search) without this restriction. Interestingly, even when it is allowed to use the search engine, the performance shows only a modest improvement (26% vs 31%), suggesting that current agents still struggle to handle real user tasks on the real-world websites. The small improvement also highlights that Online-Mind2Web tasks are less likely to be solved by shortcuts, unlike WebVoyager's tasks, 50% of which can be completed using such shortcuts.

## B.3 The influence of threshold $\theta$ of Webjudge.

Different thresholds $\theta$ (i.e., screenshots with a relevance score of task completion above or equal to a threshold $\theta$ are filtered as key screenshots) will result in different inputs for the

| Agent | Success Rate |
|---|---|
| Browser-Use (w/o Google Search) | 26% |
| Browser-Use (w/ Google Search) | 31% |

Table B.3: The performance of the Browser-Use agent with and without search constraints.

final outcome judgement. Specifically, a larger $\theta$ will significantly reduce the number of screenshots but also lose a lot of information for judging. In contrast, smaller $\theta$ will maintain most of the intermediate screenshots but also increase the issue of token overhead and suppress the model's context. Therefore, we conduct experiments with different values of $\theta$ on Operator's trajectories to show the influence of threshold $\theta$. As shown in Table B.4, we observe that $\theta$=3 achieves the highest agreement with human judgments while $\theta$=2 and $\theta$=4 show a noticeable drop in agreement. $\theta$=2 introduces too many screenshots for the final outcome judgment, overwhelming the model and making it difficult to focus on the most informative ones. In contrast, $\theta$=4 ignores many important intermediate screenshots, leading to the loss of critical information. For all experiments in the paper, we used the threshold $\theta$=3 throughout. We did not set different thresholds for different agents in order to ensure consistency and general applicability.

| WebJudge | $\theta$=2 | $\theta$=3 | $\theta$=4 |
|---|---|---|---|
| Agreement | 79.3 | **81.8** | 77.0 |

Table B.4: The agreement rates of WebJudge with different $\theta$ values.

## B.4 The Robustness of WebJudge

As shown in Figure B.1, WebJudge demonstrates strong robustness, with an average standard deviation of only 1.1% across three runs, indicating low evaluation variance.

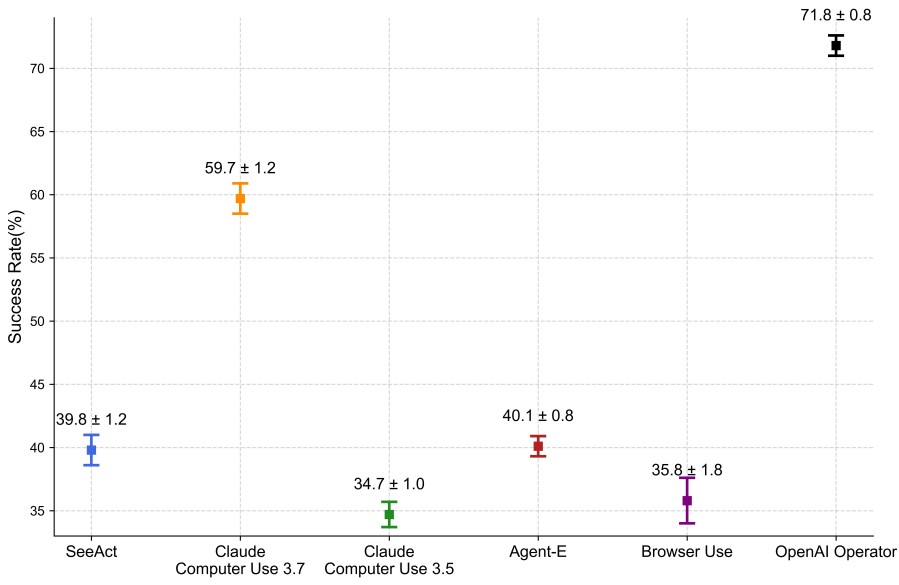

Figure B.1: Robustness of WebJudge. The Y-axis indicates the agent's success rate evaluated using WebJudge, with error bars representing the standard deviation.

### B.5 Detailed Results on AgentRewardBench

Compared to the results of GPT-4o, o4-mini demonstrates exceptionally high precision and is comparable to the rule-based method, while offering greater scalability. This high precision indicates its potential as a robust reward model for downstream applications such as reinforcement learning (RL), reinforcement tuning (RFT), and reflection. However, similar to rule-based methods, it tends to impose overly strict evaluation criteria, which can lead to relatively low recall and a larger gap when compared to human evaluation results. In contrast, WebJudge-7B achieves a balanced performance, offering high precision with a significantly improved recall rate, while reducing evaluation costs to a fixed two calls per trajectory. This makes it a reliable and scalable model for further research.

| Model | Benchmark | Precision | Recall | F1 |
|---|---|---|---|---|
| *Rule-based** | AssistantBench | 25.0 | 12.5 | 16.7 |
| | VisualWebArena | 85.2 | 58.2 | 69.2 |
| | WebArena | 79.0 | 53.8 | 64.0 |
| | WorkArena | 100.0 | 91.9 | 95.8 |
| | WorkArena++ | 83.3 | 38.5 | 52.6 |
| | Overall | 83.8 | 55.9 | 67.1 |
| o4-mini | AssistantBench | 100.0 | 25.0 | 40.0 |
| | VisualWebArena | 74.5 | 51.9 | 61.2 |
| | WebArena | 81.2 | 58.0 | 67.6 |
| | WorkArena | 100.0 | 54.1 | 70.2 |
| | WorkArena++ | 90.0 | 17.3 | 29.0 |
| | Overall | 82.0 | 47.8 | 60.4 |
| GPT-4o | AssistantBench | 66.7 | 50.0 | 57.1 |
| | VisualWebArena | 69.8 | 75.9 | 72.7 |
| | WebArena | 72.6 | 82.4 | 77.2 |
| | WorkArena | 92.3 | 64.9 | 76.2 |
| | WorkArena++ | 75.0 | 46.2 | 57.1 |
| | Overall | 73.7 | 71.2 | 72.4 |
| WebJudge-7B | AssistantBench | 80.0 | 50.0 | 61.5 |
| | VisualWebArena | 66.7 | 58.2 | 62.2 |
| | WebArena | 77.5 | 72.3 | 74.8 |
| | WorkArena | 100.0 | 56.8 | 72.4 |
| | WorkArena++ | 70.0 | 26.9 | 38.9 |
| | Overall | 75.7 | 58.0 | 65.6 |

Table B.5: Fine-grained evaluation results for WebJudge powered by GPT-4o and o4-mini.

## C Details of Task Construction

### C.1 The website selection process

For website selection, we followed a similar approach to Mind2Web, prioritizing popular websites based on traffic volume reported by SimilarWeb. Specifically, for tasks from the Mind2Web and Mind2Web-Live datasets, we chose tasks from different websites as much as possible. For the newly constructed tasks by us, we selected previously unrepresented but popular websites, ranked by traffic volume, according to SimilarWeb. To further ensure diversity, we also included 14 tasks from niche websites and several websites from different regions, such as those in the UK. Based on this construction process, we ultimately obtained 300 diverse tasks covering 136 unique websites.

## C.2 Task Distribution

As described in Sec. 2.2, our tasks are sourced from 136 popular websites spanning diverse domains, including clothing, food, housing, finance, entertainment, and transportation. We also categorize tasks into three levels of difficulty based on the number of steps (reference length). The distributions of popularity and reference length are shown in Figure C.1.

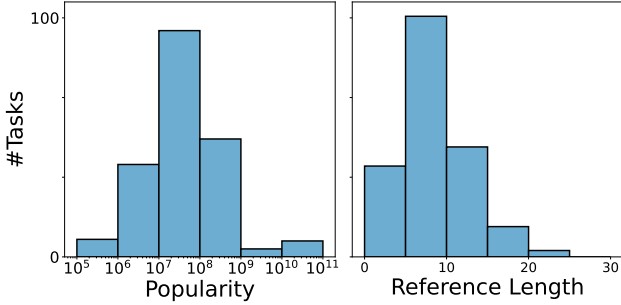

Figure C.1: The distribution of tasks on Online-Mind2Web, with respect to popularity and reference length. The popularity of a website is quantified by the monthly number of user clicks according to SimilarWeb.

We also categorize tasks into 12 domains, including Shopping & E-Commerce, Food & Recipes, Housing & Real Estate, Finance & Investment, Health & Medical, Travel & Transportation, Entertainment & Media, Technology, Education, Government & Services, Jobs & Careers, and Other. The Figure C.2 shows the number of websites for each domain.

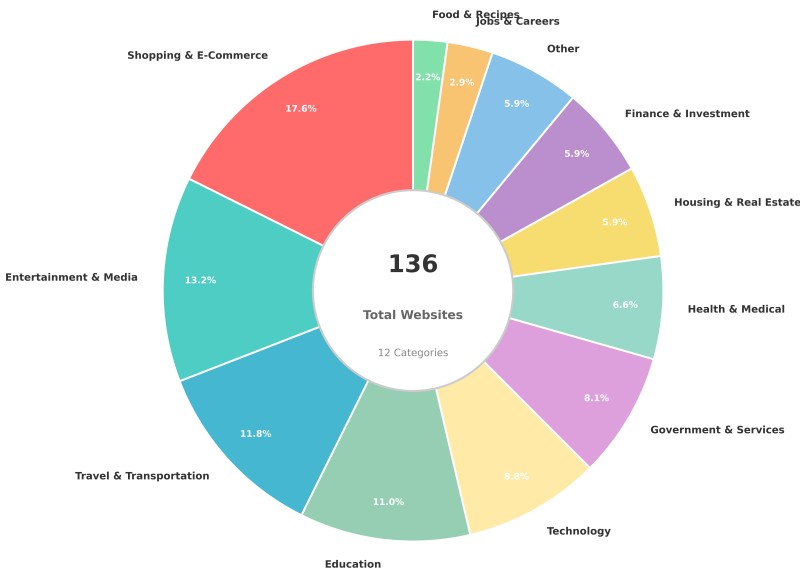

Figure C.2: The statistics distribution of task domains.

# D Prompts for WebJudge

## D.1 Online-Mind2Web-Specific Prompts

### WebJudge - Key Point Identification

You are an expert tasked with analyzing a given task to identify the key points explicitly stated in the task description.

**Objective**: Carefully analyze the task description and extract the critical elements explicitly mentioned in the task for achieving its goal.

**Instructions**:
1. Read the task description carefully.
2. Identify and extract **key points** directly stated in the task description.
- A **key point** is a critical element, condition, or step explicitly mentioned in the task description.
- Do not infer or add any unstated elements.
- Words such as "best," "highest," "cheapest," "latest," "most recent," "lowest," "closest," "highest-rated," "largest," and "newest" must go through the sort function (e.g., the key point should be "Filter by highest").
**Respond with**:
- **Key Points**: A numbered list of the explicit key points for completing this task, one per line, without explanations or additional details.

Task: {task}

### WebJudge - Key Screenshot Identification

You are an expert evaluator tasked with determining whether an image contains information about the necessary steps to complete a task.

**Objective**: Analyze the provided image and decide if it shows essential steps or evidence required for completing the task. Use your reasoning to explain your decision before assigning a score.

**Instructions**:
1. Provide a detailed description of the image, including its contents, visible elements, text (if any), and any notable features.
2. Carefully examine the image and evaluate whether it contains necessary steps or evidence crucial to task completion:
- Identify key points that could be relevant to task completion, such as actions, progress indicators, tool usage, applied filters, or step-by-step instructions.
- Does the image show actions, progress indicators, or critical information directly related to completing the task?
- Is this information indispensable for understanding or ensuring task success?
- If the image contains partial but relevant information, consider its usefulness rather than dismissing it outright.

3. Provide your response in the following format:
- **Reasoning**: [Your explanation]
- **Score**: [1-5]

**Task**: {task}

**Key Points for Task Completion**: {key points}

The snapshot of the web page is shown in the image.

**WebJudge - Outcome Judgement**

You are an expert in evaluating the performance of a web navigation agent. The agent is designed to help a human user navigate a website to complete a task. Given the user's task, the agent's action history, key points for task completion, some potentially important web pages in the agent's trajectory and their reasons, your goal is to determine whether the agent has completed the task and achieved all requirements.

Your response must strictly follow the following evaluation criteria!

**\*Important Evaluation Criteria\*:**
1: The filtered results must be displayed correctly. If filters were not properly applied (i.e., missing selection, missing confirmation, or no visible effect in results), the task is not considered successful.
2: You must carefully check whether these snapshots and action history meet these key points. Ensure that specific filter conditions, such as "best," "highest," "cheapest," "latest," "most recent," "lowest," "closest," "highest-rated," "largest," and "newest" are correctly applied using the filter function (e.g., sort function).
3: Certain key points or requirements should be applied by the filter. Otherwise, a search with all requirements as input will be deemed a failure since it cannot guarantee that all results meet the requirements!
4: If the task requires filtering by a specific range of money, years, or the number of beds and bathrooms, the applied filter must exactly match the given requirement. Any deviation results in failure. To ensure the task is successful, the applied filter must precisely match the specified range without being too broad or too narrow.
Examples of Failure Cases:
- If the requirement is less than $50, but the applied filter is less than $25, it is a failure.
- If the requirement is $1500-$2500, but the applied filter is $2000-$2500, it is a failure.
- If the requirement is $25-$200, but the applied filter is $0-$200, it is a failure.
- If the required years are 2004-2012, but the filter applied is 2001-2012, it is a failure.
- If the required years are before 2015, but the applied filter is 2000-2014, it is a failure.
- If the task requires exactly 2 beds, but the filter applied is 2+ beds, it is a failure.
5: Some tasks require a submission action or a display of results to be considered successful.
6: If the retrieved information is invalid or empty (e.g., No match was found), but the agent has correctly performed the required action, it should still be considered successful.
7: If the current page already displays all available items, then applying a filter is not necessary. As long as the agent selects items that meet the requirements (e.g., the cheapest or lowest price), the task is still considered successful.

**\*IMPORTANT\***
Format your response into two lines as shown below:

**Thoughts:** <your thoughts and reasoning process based on double-checking each key points and the evaluation criteria>
**Status:** "success" or "failure"

User Task: {task}

Key Points: {key points}

Action History: {action history}

The potentially important snapshots of the webpage in the agent's trajectory and their reasons: {thoughts}

## D.2 General-Purpose Prompt

---

**WebJudge - Key Point Identification**

You are an expert tasked with analyzing a given task to identify the key points explicitly stated in the task description.

**Objective**: Carefully analyze the task description and extract the critical elements explicitly mentioned in the task for achieving its goal.

**Instructions**:
1. Read the task description carefully.
2. Identify and extract **key points** directly stated in the task description.
- A **key point** is a critical element, condition, or step explicitly mentioned in the task description.
- Do not infer or add any unstated elements.
- Words such as "best," "highest," "cheapest," "latest," "most recent," "lowest," "closest," "highest-rated," "largest," and "newest" must go through the sort function (e.g., the key point should be "Filter by highest").
**Respond with**:
- **Key Points**: A numbered list of the explicit key points for completing this task, one per line, without explanations or additional details.

Task: {task}

---

**WebJudge - Key Screenshot Identification**

You are an expert evaluator tasked with determining whether an image contains information about the necessary steps to complete a task.

**Objective**: Analyze the provided image and decide if it shows essential steps or evidence required for completing the task. Use your reasoning to explain your decision before assigning a score.

**Instructions**:
1. Provide a detailed description of the image, including its contents, visible elements, text (if any), and any notable features.
2. Carefully examine the image and evaluate whether it contains necessary steps or evidence crucial to task completion:
- Identify key points that could be relevant to task completion, such as actions, progress indicators, tool usage, applied filters, or step-by-step instructions.
- Does the image show actions, progress indicators, or critical information directly related to completing the task?
- Is this information indispensable for understanding or ensuring task success?
- If the image contains partial but relevant information, consider its usefulness rather than dismissing it outright.

3. Provide your response in the following format:
### Reasoning: [Your explanation]
### Score: [1-5]

**Task**: {task}

**Key Points for Task Completion**: {key points}

The snapshot of the web page is shown in the image.

---

**WebJudge - Outcome Judgement**

You are an expert in evaluating the performance of a web navigation agent. The agent is designed to help a human user navigate a website to complete a task. Given the user's task, the agent's action history, key points for task completion, some potentially important web pages in the agent's trajectory and their reasons, your goal is to determine whether the agent has completed the task and achieved all requirements.

Your response must strictly follow the following evaluation criteria!

**\*Important Evaluation Criteria\*:**
1: The filtered results must be displayed correctly. If filters were not properly applied (i.e., missing selection, missing confirmation, or no visible effect in results), it should be considered a failure.
2: You must carefully check whether these snapshots and action history meet these key points. Ensure that specific filter conditions, such as "best," "highest," "cheapest," "latest," "most recent," "lowest," "closest," "highest-rated," "largest," and "newest" are correctly applied using the filter function (e.g., sort function).
3: Certain key points or requirements should be applied by the filter. Otherwise, a search with all requirements as input will be deemed a failure since it cannot guarantee that all results meet the requirements!
4: If the task requires filtering by a specific range of money, years, or the number of beds and bathrooms, the applied filter must exactly match the given requirement. Any deviation results in failure. To ensure the task is successful, the applied filter must precisely match the specified range without being too broad or too narrow.
5: Some tasks require a submission action or a display of results to be considered successful. Repeat actions or actions that do not lead to a visible result should be considered a failure.
6: If the agent loops through a sequence of actions that do not make progress toward the goal (including failing to click "Save" or "Submit," etc.), it should be considered a failure.

Format your response into two lines as shown below:
**Thoughts:** <your thoughts and reasoning process based on double-checking each key points and the evaluation criteria>
**Status:** "success" or "failure"

User Task: {task}

Key Points: {key points}

Action History: {action history}

The potentially important snapshots of the webpage in the agent's trajectory and their reasons: {thoughts}

# E   Implementation Details

**Software Tools and Libraries:** The open-source agents, including SeeAct, Browser Use, and Agent-E, are evaluated online using Playwright, with a maximum step limit of 25 to control the cost of repeated actions. For Claude Computer Use, we utilize the Computer Use OOTB Tool (Hu et al., 2024) to conduct tests on a local Chrome browser. To reduce cost, we set the maximum step limit to 50 for Claude Computer Use 3.7 and disable the thinking mode. Operator can only be run on the remote browser provided by OpenAI and does not provide API access, so we collect actions and screenshots from Operator's web-based interface for evaluation.

**Base Model and Configuration:** We employ gpt-4o-2024-08-06 as the backbone for SeeAct, Browser Use and Agent-E, claude-3-5-sonnet-20241022 for Claude Computer Use. For automatic evaluators, we use gpt-4o-2024-08-06 and o4-mini-2025-04-16 as the base models. We set the temperature of GPT-4o to 0 and use the default reasoning effort level of medium for o4-mini. The threshold $\delta$ of WebJudge is set to 3.

**Agent Trajectories:** Different web agents adopt different viewpoints in web interaction. Specifically, SeeAct and Agent-E are designed to capture extended full-page screenshots, whereas Browser Use, Claude Computer Use and Operator captures only the visible portion of the screen.

**Action History:** For the action sequence of SeeAct, Browser Use and Agent-E, we first filter specific attributes of elements and then combine them with the corresponding actions (e.g., `CLICK, TYPE`), resulting in a unified action representation such as `<aria-label="Email"> -> TYPE myemail@gmail.com`. For Claude Computer Use, we also incorporate click coordinates into the action representation, leveraging the inherent grounding capabilities present in many existing models. For Operator, we directly use the provided action descriptions as their representations.

**AgentRewardBench Evaluation** We use gpt-4o-2024-11-20 as the backbone to ensure consistency with prior results. For the action history, we concatenate each action with its corresponding reasoning as a single action. We rely solely on screenshots rather than A11Y trees, since some agents (e.g., Operator) do not provide A11Y content and not all websites support it. Moreover, incorporating A11Y trees introduces additional input, which significantly increases both latency and inference costs (Gou et al., 2025). We use the general-purpose WebJudge prompt, with only minimal modifications to the prompt that was previously tailored for Online-Mind2Web tasks. For WebJudge-7B, we maintain consistency by using GPT-4o-2024-11-20 for both key point identification and outcome judgment stages.

**WebJudge-7B** We synthesize training data from the key screenshot identification stages of several models, including GPT-4o, GPT-4.1-mini, Claude 3.7, and Qwen2.5-VL-72B. We use trajectories from SeeAct, Browser Use, and Claude Computer Use 3.5 for training, and reserve Agent-E, Claude Computer Use 3.7, and Operator as held-out evaluation sets. During inference, the trained model summarizes each screenshot and assigns a relevance score, while key point identification and outcome judgment are still handled by an LLM (e.g., GPT-4o and o4-mini). This reduces the number of API calls from being proportional to the trajectory length to a fixed two calls per trajectory. We fine-tune WebJudge-7B on 4×H100 (80GB) GPUs using bfloat16 precision to reduce memory. Training is conducted with a batch size of 256 and a maximum sequence length of 4096 over 5 epochs. We use a learning rate of 5e-6 with a cosine learning rate scheduler and a warmup ratio of 10%. During inference, key point identification and outcome judgment stages are powered by o4-mini.

# F   Case Study

## F.1   Case Study: Operator

This section presents case studies demonstrating the strengths of Operator in completing complex web tasks.

### F.1.1   *Effective Utilization of Structured Search*

Figure F.1 is an example of Operator applying filters to complete the task.

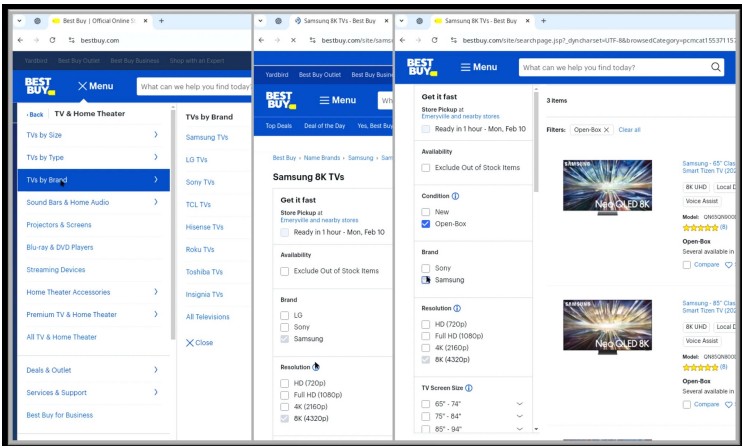

Figure F.1: Task: "Browse 8K Samsung TVs that are open box."

### F.1.2   *Leveraging Tool-Based Navigation*

Figure F.2 is an example of Operator using the "Ctrl+F" tool. Operator searches the page for the keyword "Compare" and quickly identifies the "Compare Side by Side" feature.

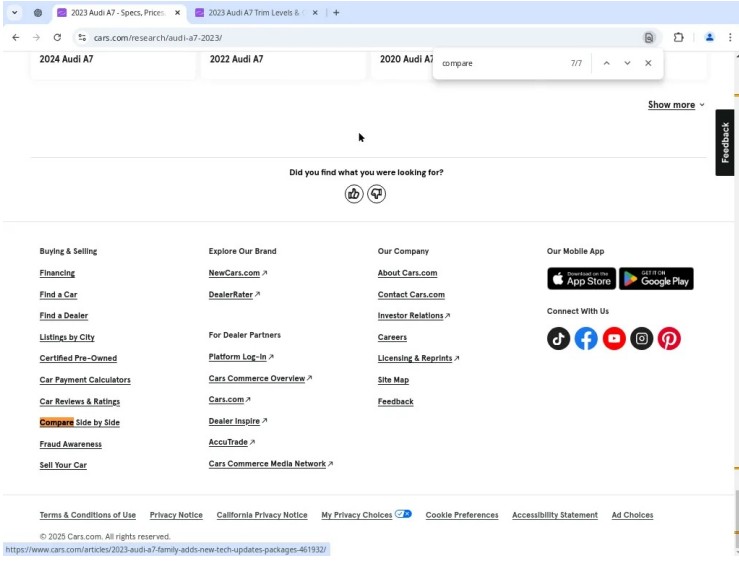

Figure F.2: Task: "Compare Audi A7 with Audi A6 both made in 2023 and hide similarities"

### F.1.3  Self-Verification and Error Correction

Figure F.3 is an example of Operator conducting self-verification and self-correction. Given the task "Show me the list of Men's Blazers, Black, Size M on uniqlo.", Operator intends to select BLACK, but due to a grounding error, mistakenly chooses BLUE. It then identifies and corrects the mistake autonomously.

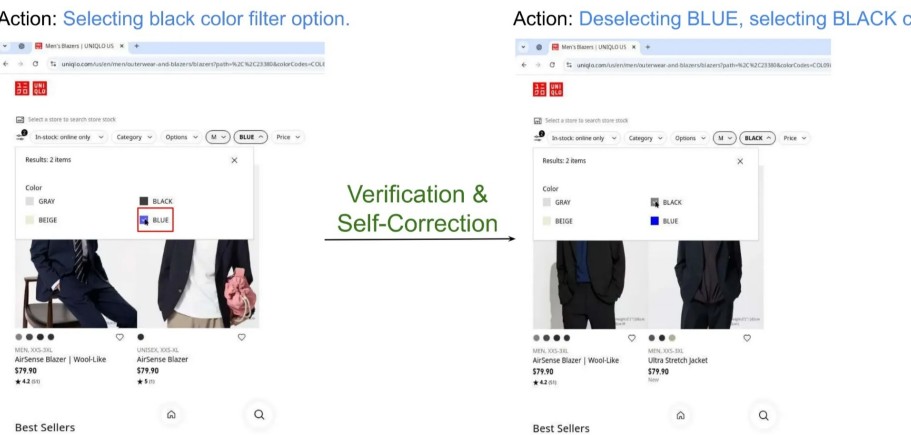

Figure F.3: Task: "Show me the list of Men's Blazers, Black, Size M on uniqlo."

### F.1.4  Failure Cases about Numeric and Temporal Constraints

Figure F.4 is an example of Operator applying an incorrect broader time range of 2001 to 2012 instead of the specified 2004 to 2012.

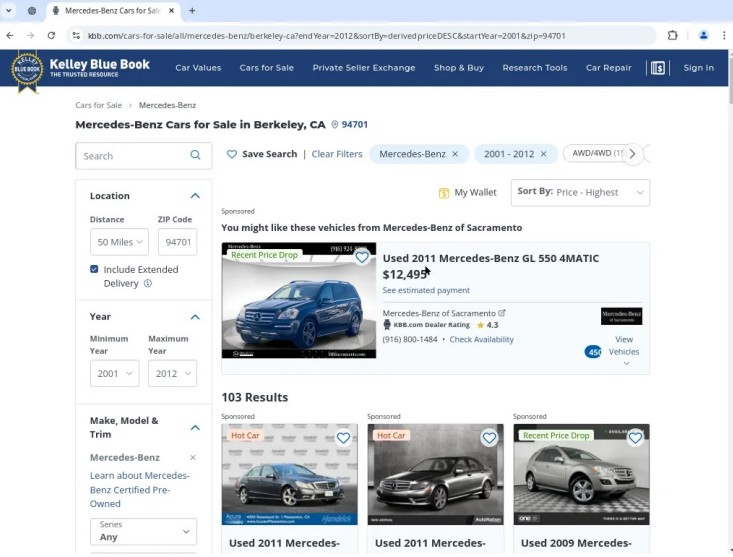

Figure F.4: Task: "Browse used Mercedes-Benz cars from model years 2004 to 2012 on KBB and sort by highest price."

Figure F.5 is an example of Operator failing to adjust the time slider correctly.

### F.2  Case Study: Other Agents

Figure F.2 is an example of over-reliance on hasty keyword search: all three agents issue a single, loosely structured query.

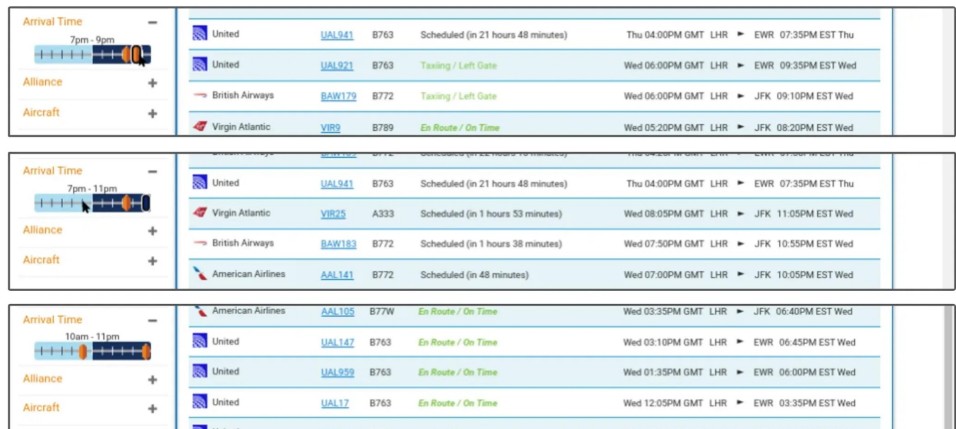

Figure F.5: Task: "Find UA or AA flights from London to New York that arrive between 8:00 PM and 11:00 PM on FlightAware."

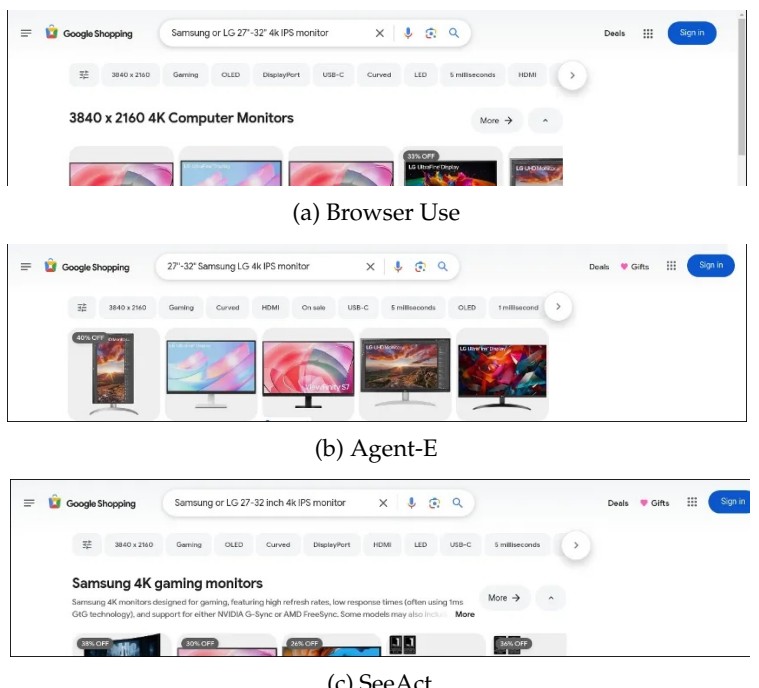

(a) Browser Use

(b) Agent-E

(c) SeeAct

Figure F.6: Task: "Find the lowest-priced 27"–32" Samsung or LG computer monitors with a 4K IPS display."

### F.3 Examples of Hallucinations

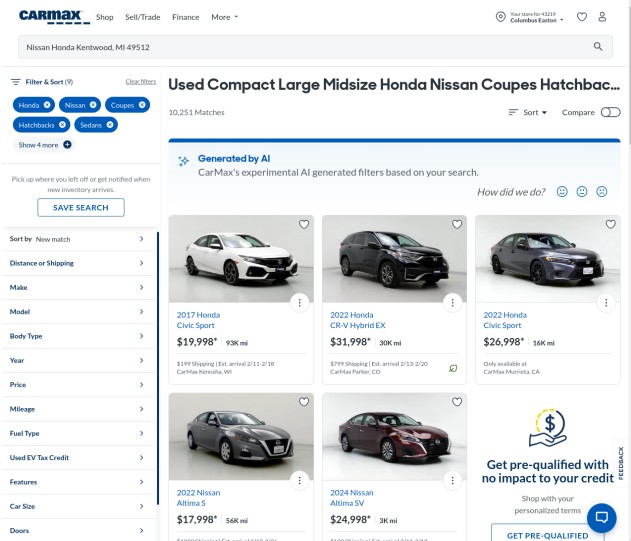

Figure F.7: Task: "See Nissan and Honda cars for sale near Kentwood, MI 49512 on CarMax."

The final response from Browser Use is *"Here are some Nissan and Honda cars for sale near Kentwood, MI 49512 on CarMax:1. **2017 Honda Civic Sport** - Price: $19,998 - Mileage: 93,000 miles CarMax Kenosha, WI - Shipping: $199. **2022 Honda Civic Sport** - Price: $26,998 - Mileage: 16,000 miles - Location: CarMax Murrieta, CA ......"* However, as shown in Figure F.7, these cars are not located near Kentwood, MI 49512, indicating a typical case of hallucinated content.

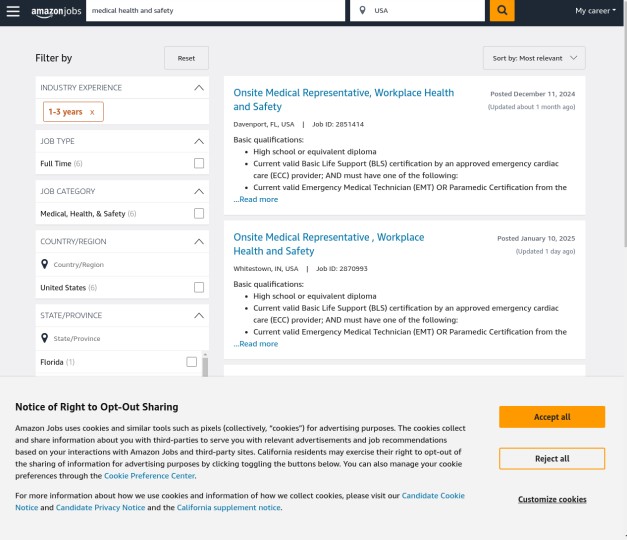

Figure F.8: Task: "Check the most recent full-time medical health and safety jobs, requiring 1-3 years of industry experience available in the US."

The final response from Browser Use is *"Here are the most recent full-time medical health and safety jobs in the US requiring 1-3 years of industry experience: 1. **Onsite Medical Representative, Workplace Health and Safety ......"* However, as shown in Figure F.8, the agent does not utilize the sort function but instead directly claims that the listed job is the most recent.

# G  Task Examples

| Websites | Task Description |
|---|---|
| **Mind2Web-Live** | |
| *https://www.carmax.com/* | Find a 2022 Tesla Model 3 on CarMax. |
| *https://www.qatarairways.com/* | Find the weight of baggage allowance for economy class on Qatar Airways. |
| *https://www.kbb.com/* | Browse used Audi cars made before 2015 and sort by lowest price on KBB. |
| *https://us.megabus.com/* | Find out what to do when I lose an item on a bus on us.megabus. |
| *https://www.amtrak.com/* | Tell me information about what identification I need to bring on my trip on Amtrak. |
| **Mind2Web** | |
| *https://www.fedex.com/* | Calculate a FedEx Ground shipping rate for a 3-pound package from zip code 10019 to zip code 90028. |
| *https://www.macys.com/* | Search for boys' infant pajamas below $40. |
| *https://www.healthgrades.com/* | Browse dermatologists within 10 miles of zip code 10019 and filter by only those who accept Blue Medicare Advantage |
| *https://www.redfin.com/* | Find a premier real estate agent in St Augustine, FL. |
| *https://www.student.com/* | Show me the shared rooms in any university in Melbourne that has a private bathroom wifi, and gas included in the bills |
| **Online-Mind2Web** | |
| *https://soundcloud.com* | Browse a user homepage that reposted the top song from the Top 50 Rock chart. |
| *https://github.com/* | Identify the open issue with the most comments in the first trending open-source repository this week. |
| *https://iclr.cc/* | Open the page for the first Best Paper Award video recording of talks from ICLR 2016. |
| *https://www.imdb.com/* | Browse the top 250 movies and find one movie that is available on AMC+. |
| *https://www.google.com/maps* | Find the top-rated hotel in Manhattan, NY, suitable for 4 guests, and identify the fastest public transportation option from the hotel to LGA airport. |
| *https://smartasset.com/* | Estimate the federal income tax I would owe on $158,500 of taxable income in ZIP code 97007, filing as single. |
| *https://imgur.com/* | Create a meme with a frog as the background and leave the only text with "Enjoy your life". |
| *https://www.chess.com/* | Pass the first trending chess puzzle. |
| *https://www.nvidia.com/* | Find the HGX H100 driver for Ubuntu 22.04 on AMD64 CPU. |
| *https://www.berkeley.edu/* | Please find graduate-level computer science courses scheduled on Tuesdays starting time from 2:00 to 6:00 PM in the Fall 2023 semester. |

Table G.1: Task examples from Online-Mind2Web benchmark. This table showcases 20 tasks sampled from our datasets: 5 from Mind2Web-Live, 5 from the original Mind2Web, and 10 from the newly constructed tasks. The new tasks cover a wider range of domains, including academia, entertainment, transportation, and finance, and introduce more realistic scenarios such as creating memes and planning travel using Google Maps.

## H  Examples of WebJudge

**Task:** Find a male senior boxer near zip code 90028.

**Key Points:**
1. Breed: boxer
2. Gender: male
3. Age category: senior
4. Location: near zip code 90028

**Key Screenshots Identification:**
**Screenshot 1**
Reasoning: The image shows only a blank Chrome new tab page with an empty address/search bar and no visible content, filters, search results, or any information about locating a male senior boxer near zip code 90028. There are no steps, no filter settings, no listings, and no location data displayed. Therefore it provides no relevant evidence for completing the task. **Score: 1**
......
**Screenshot M**
Reasoning: The screenshot displays a pet search form with Breed: Boxer and Age: Senior filters already applied, and it shows search results including Bingo Bongo and Peanut, both male senior Boxers within the specified 50-mile radius of ZIP code 90028. However, the Sex filter remains unchecked ...... This makes the information highly relevant but not fully comprehensive for completing the task. **Score: 4**

**Outcome Judgment:**
Thoughts: The agent correctly set location (90028), age (Senior), and breed (Boxer), then viewed the full set of six available Senior Boxers within 50 miles. From that complete list it identified Bingo Bongo, a male senior Boxer, and provided his details meeting all four criteria even though it did not explicitly toggle the gender filter. By displaying and choosing from the full set of results, the agent ensures the selection satisfies location, breed, age, and gender. **Status: success**

