# OpenReview forum: "An Illusion of Progress? Assessing the Current State of Web Agents"
_colmweb.org/COLM/2025/Conference — COLM 2025_

### Official Review · Reviewer_8d2e · 2025-05-11

**Rating:** 8
**Confidence:** 4
**Ethics Flag:** 1

**Summary:**

The paper starts by analyzing the evaluations of existing web agents on popular benchmarks, criticizing the over-estimation of their capabilities by identifying significant shortcomings in the evaluation methodologies. The authors follow the analysis by proposing a more realistic and challenging online benchmark comprising 300 tasks across 136 live websites (which, per their claims, would be maintained) called Online-Mind2Web. An analysis similar to the one performed on the existing benchmarks shows that Online-Mind2Web is more robust and presents a more logical progression in web agents' research, especially given the "hard" set of tasks. The authors also propose an interesting LLM-as-a-judge-based method to evaluate the performance of popular baselines on their benchmark, obtaining significantly higher agreement with human evaluators. Overall, I believe the contributions made in the paper are extensive and can be important resources for progressing in this research direction.

**Questions To Authors:**

1. How do the authors plan to maintain the Online-Mind2Web benchmark given the dynamic nature of web environments? Can the authors clarify the practical scalability of their proposed maintenance strategy?
2. Can the authors elaborate on the criteria for selecting the five agents evaluated? Were there additional agents considered but excluded?
3. Not a question, but adding qualitative examples from the WebJudge evaluation in the appendix would help readers better understanding the process.

**Reasons To Accept:**

1. The analytical methodology of the authors when critiquing existing benchmarks is sound, enhancing the confidence in their conclusions.
2. The two main resource contributions of the work apart from the analysis: the benchmark and the WebJudge method, would prove to be important for furthering research for GUI agents.
3. Given constraints of paper length, the error analysis is detailed, interesting, and thorough.

**Reasons To Reject:**

1. The scope of evaluation is limited in terms of agents/agent frameworks used. It would be interesting to see the results of open-source models like UI-TARS.
2. Similarly, given that closed-source models like GPT-4o do not remain consistent over time and open-source models are getting competitive, it will be nice if the WebJudge method is used in conjunction with a strong open-source model to verify their agreement.
2. Though the authors claim that they would maintain the benchmark over time, realistically, continuous maintenance required for the Online-Mind2Web benchmark would be challenging.

---

> ### Author Response · Authors · 2025-06-01
> **Author Response to Reviewer 8d2e (Part 1/2)**
>
> We appreciate that the reviewer recognizes the thorough error analysis and the significance of the benchmark and WebJudge method in advancing research on GUI agents. We thank the reviewer for their detailed feedback and constructive suggestions. Here, we treasure the opportunity to address your concerns and improve the quality of our work.
>
> ---
>
> **W1: The results of open-source models like UI-TARS.**
>
> Thanks for pointing this out. The results for open-source models are indeed very important, as they help identify shortcomings and guide future improvements. We find that UI-TARS(UI-TARS-1.5-7B) has already reported its performance on Online-Mind2Web[1]. It achieves a 75.8% success rate evaluated by WebJudge (GPT-4o), which even surpasses Operator. It’s also worth noting that UI-TARS-1.5-7B has achieved new state-of-the-art performance on several other benchmarks, such as OSWorld and AndroidWorld, demonstrating that it is indeed a very strong open-source agent model. We have been in contact with the UI-TARS team to validate their reported results.
>
> ---
>
> **W2: It will be nice if the WebJudge method can based on open-source model.**
>
> Thank you for the great suggestion! We synthesized training data from multiple models and trained a lightweight yet reliable WebJudge-7B, based on Qwen2.5-VL-7B. This model achieves a high agreement rate with human judgments (87%) and exhibits a small success rate gap of just 3.9% compared to human evaluation. Moreover, it generalizes well to out-of-distribution (OOD) benchmarks. Please refer to our general response for more details.
>
> ---
>
> **W3 and Q1: The maintenance of Online-Mind2Web.**
>
> We acknowledge that maintaining the leaderboard is challenging, but we are committed to putting in significant effort to ensure its long-term value. Please see our general response for more details.
>
> ---
>
> **Q2: The selection of agents.**
>
> This is a great question. In general, we selected well-known and widely recognized agents, especially those that claim to achieve very high success rates on WebVoyager, such as Browser-Use, to rigorously evaluate whether their performance still holds in real-world scenarios. We conducted a thorough literature review [2-4] and excluded agents that could not be evaluated (e.g., those that are not open-source or lack API acess).  We ended up with the five agents for evaluation. We also considered including Google DeepMind's Project Mariner, but as of the submission time, we were unable to access its internal API, so it could not be included.
>
> We want to note that we also evaluated Claude Computer Use 3.7 recently, a stronger and highly anticipated agent from Anthropic. Compared to version 3.5, it shows notable improvements, with stronger exploration capabilities and a tendency to produce longer trajectories, similar to Operator. However, it still falls short of Operator, achieving only a 56.3% (under human evaluation) success rate and struggling with more complex tasks.

---

> > ### Author Response · Authors · 2025-06-01
> > **Author Response to Reviewer 8d2e (Part 2/2)**
> >
> > **Q3: Qualitative examples for WebJudge.**
> >
> > Thanks for the suggestion! We will add some examples in the revision to better help readers understand the process of WebJudge. Here is an example showing key parts of WebJudge.
> >
> > **Task**: Find a male senior boxer near zip code 90028.
> >
> > **Key Points:**
> > 1. Breed: boxer
> > 2. Gender: male
> > 3. Age category: senior
> > 4. Location: near zip code 90028
> >
> > **Key Screenshots Identification**:
> >
> > **Screenshot 1**
> >
> > **Reasoning**: The image shows only a blank Chrome new tab page with an empty address/search bar and no visible content, filters, search results, or any information about locating a male senior boxer near zip code 90028. There are no steps, no filter settings, no listings, and no location data displayed. Therefore it provides no relevant evidence for completing the task.
> > **Score: 1**
> >
> > ……
> >
> > **Screenshot M**
> >
> > **Reasoning**: The screenshot displays a pet search form with Breed: Boxer and Age: Senior filters already applied, and it shows search results including Bingo Bongo and Peanut, both male senior Boxers within the specified 50-mile radius of ZIP code 90028. However, the Sex filter remains unchecked …… This makes the information highly relevant but not fully comprehensive for completing the task.
> > **Score: 4**
> >
> > **Outcome Judgment:**
> >
> > Thoughts: The agent correctly set location (90028), age (Senior), and breed (Boxer), then viewed the full set of six available Senior Boxers within 50 miles. From that complete list it identified Bingo Bongo, a male senior Boxer, and provided his details meeting all four criteria even though it did not explicitly toggle the gender filter. By displaying and choosing from the full set of results, the agent ensures the selection satisfies location, breed, age, and gender.
> > **Status: success**
> >
> > We also provide a brief summary of the process of WebJudge to make it clearer. WebJudge consists of three main components: key point identification, key node screenshot selection, and outcome judgment. In the first step, the model identifies the essential key points required to complete the task. In the second step, it retains important intermediate screenshots while filtering out irrelevant frames. Finally, the evaluator determines whether the task was successfully completed or not.
> >
> > **Reference:**
> >
> > >[1] https://seed-tars.com/1.5/
> >
> > >[2] https://browser-use.com/posts/sota-technical-report
> >
> > >[3] https://www.hcompany.ai/blog/a-research-update
> >
> > >[4] https://openai.com/index/introducing-operator

---

> > > ### Author Response · Authors · 2025-06-08
> > > **Gentle Reminder from Authors**
> > >
> > > Dear Reviewer 8d2e,
> > >
> > > As the end of discussion period is approaching, we would like to gently remind you of our responses to your comments. We wonder whether your concerns have been addressed and appreciate any further questions or comments you might have.
> > >
> > > Sincerely,
> > >
> > > Authors of Submission 1009

---

> > > > ### Comment · Reviewer_8d2e · 2025-06-10
> > > >
> > > > Thank you for your comments and the extensive rebuttal. I am satisfied with the additional details provided in the general response and the specific reviews and have updated my score to reflect the same. Good luck.

---

> > > > > ### Author Response · Authors · 2025-06-11
> > > > >
> > > > > Thank you for the great discussions! We will integrate them into the revised version. We sincerely appreciate your recognition of our efforts.

---

### Official Review · Reviewer_BGA6 · 2025-05-12

**Rating:** 6
**Confidence:** 2
**Ethics Flag:** 1

**Summary:**

This paper presents a novel evaluation system for LLM based Web agents which seeks to be more realistic in its results than previous ones.

**Questions To Authors:**

Will the dataset and code be realeased, under a free software licence? I think I did not read that in the paper.

Is it really fair to prevent agents to use a search engine (or any? you talk only about Google)? Even if the answer is present in 50% of the search results, it is just a natural way to solve a task on the Web.




Questions:
What is a MLLM?
What is a "frontier web agent"?
Fig 2 cannot be read in B&W

**Reasons To Accept:**

The paper is clear and fine to read. Globally, the comparison with other systems and the description of experiments are clear and detailed.

**Reasons To Reject:**

The paper is globally clear except that it does not adequately presents the subject. If the reader does not know Web agents, it is difficult for him/her to understand precisely what is a task. Is it a simple Web search query or another thing? And screenshots, are they gloab Web page image screenshots or the text (HTML) and possibly associated images. The context and the only example allows to understand that tasks are complex queries that can necessitate to accesse and compare Web pages. For screenshots, I still am not sure.

The main criticism, but that applies to previous works too, rely on the fundamental evolutive state of teh Web which makes the static dataset fundamentally always outdated. I acknowlege the willing of authors to maintain and update their work, but such a maintainance always finishes. It would be necessary, but it is another research subject, to be able to automatically update this kind of dataset.

---

> ### Author Response · Authors · 2025-06-01
> **Author Response to Reviewer BGA6**
>
> We sincerely appreciate the reviewer’s recognition of our contributions and acknowledgment that our experiments are clear and detailed. We address your questions in the following texts:
>
> ---
>
> **W1: "The paper is globally clear except that it does not adequately presents the subject."**
>
> Thanks for sharing this concern. In Appendix Table G.1, we provide a variety of example tasks from Online-Mind2Web, which can help readers who do not know web agents better understand our benchmark. Overall, our tasks are more diverse and closely aligned with real-world user needs. Completing these tasks requires web agents to browse and interact with websites through a series of actions, such as clicking and typing. Our benchmark includes not only information-seeking tasks (i.e., where agents interact with the web to find specific answers) but also contains tasks that require agents to perform real operations, such as adding items to a cart or modifying a delivery address. Regarding screenshots, we refer to web page image screenshots, not HTML-based representations. Thank you for pointing this out. We will clarify this further in the revision.
>
> ---
>
> **W2: "The static dataset fundamentally always outdated".**
>
> Thanks for bringing up this important issue. Indeed, updating tasks over time is both important and challenging. We already have a plan for updating tasks and maintaining the benchmark to ensure it remains valuable in the long term. Please refer to our general response for more details.
>
> ---
>
> **Q1: Release code and dataset.**
>
> We will release all our results, models, datasets, and code under a free software license. We will add it in the revision to make it clearer.
>
> ---
>
> **Q2: No Google Search.**
>
> We acknowledge that employing search engines is a natural way to solve a task. To better understand its impact, we evaluate the Browser-Use agent without this restriction. Interestingly, even when it is allowed to use Google Search, its performance did not improve significantly(26% vs 31%), suggesting that current agents still struggle to handle real-world tasks on real-world websites. Please see our general response for more details.
>
> ---
>
> **Q3: "What is MLLM? What is a "frontier web agent? Fig 2 cannot be read in B&W."**
>
> We appreciate your feedback. We will further clarify these terms in the revision. "MLLM" refers to Multimodal Large Language Model. "Frontier web agent" refers to current state-of-the-art and well-known web agents, such as OpenAI's Operator, Anthropic's Claude Computer Use, and open-source agents like Browser-Use. We will revise Figure 2 to make it readable for black-and-white print.

---

> > ### Comment · Reviewer_BGA6 · 2025-06-05
> >
> > Thanks for your answers. I maintain my ratings.

---

> > > ### Author Response · Authors · 2025-06-07
> > >
> > > Thank you for your response. We are glad to have addressed your concerns. We will clarify these in our revision. If you have any further questions, we'd be happy to address them.

---

### Official Review · Reviewer_9u2C · 2025-05-13

**Rating:** 7
**Confidence:** 4
**Ethics Flag:** 1

**Summary:**

This paper investigates the real-world effectiveness of Web Agents, highlighting how the seemingly high performance of current agents may stem from limitations in existing benchmarks and evaluation systems. To address these issues, the authors introduce Online-Mind2Web, a new benchmark with 300 tasks across 136 live websites, categorized by difficulty, and propose WebJudge, an LLM-as-a-judge based automatic evaluation method showing strong agreement with human evaluations. Using the proposed benchmark, they present an analysis of the performance and failure cases of 5 prominent web agents.

**Questions To Authors:**

- Based on your WebJudge experiements, do you think  it can be used to provide reliable fine-grained error evaluations? For instance, for the identified keypoints in a task, how accurately can it determine which were successfully or unsuccessfully executed in the agent's trajectory? Do you anticipate that aggregating individual labels in this manner (e.g., classifying a task evaluation as successful only if all individual keypoint labels are successful), rather than generating a single success/failure evaluation label, would cause any alterations in the agreement scores?

- Can you provide some examples for the 47% of the tasks that were deemed invalid (Line 164)?

**Reasons To Accept:**

- The paper is well-written and organized with a clear structure. The need for improved web agent evaluation benchmarks is well-motivated and supported by empirical evidence.
- The evaluations were comprehensive and robust, incorporating human annotations (two annotators per task, with a third resolving conflicts), providing a thorough and credible assessment.
- As demonstrated by the experiments, the proposed benchmark poses a significant challenge for current SoTA agents. Alongside the benchmark itself, the case studies on the failure modes of current agents can serve as a valuable resource for the community to further enhance the generalizability of current web agents in real-world tasks on live websites.

**Reasons To Reject:**

- While the proposed WebJudge evaluation method demonstrates promising agreement with human evaluations, the paper would benefit from a more in-depth meta-evaluation of its limitations. A discussion of potential failure modes or scenarios where WebJudge might deviate from human judgment would provide a more comprehensive understanding of its applicability.

---

> ### Author Response · Authors · 2025-06-01
> **Author Response to Reviewer 9u2C (Part 1/2)**
>
> We sincerely appreciate the constructive comments and thoughtful questions. We greatly appreciate your recognition of our rigorous experimentation, comprehensive analysis and effective automatic evaluation method. We will address your questions in the following paragraphs.
>
> ---
>
> **W1: The discussion of WebJudge’s limitations.**
>
> Thank you for the great suggestion! While WebJudge achieves higher agreement compared to existing methods, we acknowledge that it still has certain limitations. We have conducted a thorough analysis of 15 cases where WebJudge's evaluations deviate from human judgments and provide several insights that may inform future research on developing more reliable automatic evaluation methods.
> Among the 15 error cases, 5 involve missing task requirements despite identifying them as key points, 6 contain incorrectly generated key points, and the remaining 4 fall into other categories (e.g., assigning low scores to important screenshots, thereby missing key information, or failing to capture certain details within screenshots).
> - **Missing task requirements despite identifying them as key points.**
> In some cases, WebJudge overlooks specific task requirements in the final outcome judgment phase, even when these requirements were explicitly identified as key points. For instance, if a task specifies searching houses with exactly two beds but the applied filter is “2+” beds (i.e., 2, 3, 4, or more), WebJudge may still deem the result correct. Similarly, when a task requires finding a document from the year 2023, WebJudge may ignore this time constraint and hallucinate that all requirements have been met, despite having correctly highlighted it as a key point. We suppose that this behavior may be due to the large input context in the outcome judgment phase, which includes numerous image tokens and generated summaries, potentially causing the model to overlook previously identified key points. We will add the discussions to the revised paper.
> - **The generated key points are not always accurate.**
> The key points generated by WebJudge may not be accurate and sometimes include unnecessary constraints, resulting in overly strict evaluations. For instance, in a task that involves browsing upcoming superbike events, the key points may specify the need to apply an “upcoming” filter. However, if the website displays upcoming events by default and does not provide such a filter, WebJudge may incorrectly judge it as a failure due to the perceived absence of this filter.  This issue arises because the key point generation phase only depends on the task description, without considering the website. A promising direction for future work is to make key point generation dynamic, allowing key points to be refined as the web agent explores the environment. As the agent gains a deeper understanding of the website, this could lead to more accurate key points.
>
> ---
>
> **Q1.1: Is it possible to use WebJudge to conduct fine-grained error analysis?**
>
> This is a very interesting question! Based on our experience, WebJudge can accurately identify which key points are not satisfied. We randomly reviewed 10 tasks to evaluate the accuracy of key point assessment and found that WebJudge achieves a high accuracy of 90.3%. This suggests the potential of using WebJudge for fine-grained analysis. However, as we discussed in W1, there are still some cases (i.e., 6 out of 15 cases) where the generated key points are inaccurate.
>
> ---
>
> **Q1.2:  Aggregating individual key point labels for WebJudge.**
>
> Thank you for bringing up this interesting question! In fact, in the final outcome judgment stage of WebJudge, we explicitly prompt the model to evaluate each key point before deciding whether the task is complete(i.e., “Thoughts: your thoughts and reasoning process based on double-checking each key point and the evaluation criteria”). Therefore, in a sense, we currently require the model to verify each key point in a chain-of-thought manner before outputting the final outcome judgment (success or failure).

---

> > ### Author Response · Authors · 2025-06-08
> > **Author Response to Reviewer 9u2C (Part 2/2)**
> >
> > **Q2: Some Invalid examples in the original Mind2Web.**
> >
> > Here are several types and examples of invalid tasks:
> >
> > **Website no longer available:** For example, the tiktok.music website has been taken down, rendering all tasks associated with it invalid.
> >
> >
> > **Outdated tasks:** For instance, the task "Find and purchase two aisle seats for the Adele concert in Las Vegas on June 16th" is now outdated, as there are no scheduled Adele concerts this year.
> >
> >
> > **Website changes:** Some websites have changed in ways that make the tasks unsolvable. For example, the Glassdoor website now requires users to log in before performing any actions, making tasks unexecutable. Similarly, the task "Play a past episode of one of their podcasts on the AccuWeather website" is no longer valid, as the podcast feature has been removed.

---

> > ### Comment · Reviewer_9u2C · 2025-06-08
> >
> > Thank you for your answers. I will maintain my ratings.
> >
> > > **Missing task requirements despite identifying them as key points.**: ......"We suppose that this behavior may be due to the large input context in the outcome judgment phase, which includes numerous image tokens and generated summaries, potentially causing the model to overlook previously identified key points......"
> >
> > Doesn't this suggest evaluating each key point in isolation and aggregating those labels would further improve the accuracy and reliability of WebJudge, as the input context required for individual keypoint evaluation would be significantly lower than the current setting?
> >
> > I suppose that would increase the total token cost for evaluation as well. I would love to see a discussion with some experimental results, outlining the performance and token usage differences between these two alternatives in the final version of the paper.

---

> > > ### Author Response · Authors · 2025-06-08
> > >
> > > Thank you for your suggestion. We evaluated WebJudge (Decompose), i.e., classifying a task evaluation as successful only if all individual keypoint labels are successful. As shown in the table below, when the number of key points is 3 or less(the average number of key points is 3.6 across 300 tasks), WebJudge (Decompose) achieves a higher agreement rate of 87.9%. However, when the number of key points exceeds 3, the WebJudge(CoT) performs better. This is because the generated key points are not always accurate or necessary. Evaluating each key point individually can lead to overly strict assessments, thereby lowering the agreement rate. As we discussed in W1, enabling the agent to dynamically adjust key points while interacting with the website may yield more reliable evaluations, as it gains a deeper understanding of the task and the website structure. We leave it as future work.
> > > We also found that combining both modes can further improve the overall agreement rate to 84.6%. WebJudge (Combined) leverages the strengths of each mode while mitigating their respective weaknesses, resulting in a more reliable evaluation.
> > >
> > > **Agreement rate across different modes of WebJudge**
> > > | Agreement rates on Operator trajectories | WebJudge (CoT) | WebJudge (Decompose) | WebJudge (Combined) |
> > > |----------------------------------------------|----------------|-----------------------|----------------------|
> > > | Key Points ≤ 3                                | 86.1           | **87.9**              | -                    |
> > > | Key Points > 3                                | **81.4**       | 76.3                  | -                    |
> > > | Overall                                       | 83.7           | 81.7                  | **84.6**             |
> > >
> > > We also compared the cost of the two methods. Since WebJudge (Decompose) evaluates each key point individually, its cost during the outcome judgment phase scales proportionally with the number of key points. In contrast, WebJudge (CoT) requires only a single call, resulting in approximately half the total cost of WebJudge (Decompose). Considering the trade-off between performance and cost, we believe WebJudge (CoT) is a better choice.
> > >
> > > **Token cost comparison between different modes of WebJudge**
> > > | Average Token per Task | Key Point Identification | Key Screenshot Identification | Outcome Judgment | Total Tokens per Task |
> > > |------------------------|---------------------------|-------------------------------|------------------|------------------------|
> > > | WebJudge (CoT)         | 241                       | 47k                           | 20k              | **67k**                |
> > > | WebJudge (Decompose)   | 241                       | 47k                           | 73k              | 120k                   |
> > >
> > > Thank you again for bringing up these interesting discussions. We will incorporate them into the final version of the paper.

---

> > > > ### Comment · Reviewer_9u2C · 2025-06-09
> > > >
> > > > Thank you for your prompt response and experimental results. The findings are indeed intriguing and show scope for interesting future work.

---

> > > > > ### Author Response · Authors · 2025-06-09
> > > > >
> > > > > Thank you for your response. We are glad to have addressed your concerns and appreciate you bringing up these interesting discussions. We will add these discussions to our final revision. If you have any further questions, we'd be happy to address them.

---

### Official Review · Reviewer_inKQ · 2025-05-18

**Rating:** 6
**Confidence:** 4
**Ethics Flag:** 1

**Summary:**

The work aims to critique over-optimism in web agent evaluation by introducing the Online-Mind2Web benchmark and WebJudge evaluation method. It effectively summarizes the gap between prior benchmark results and real-world performance, as well as the new contributions.

**Questions To Authors:**

besides the problems mentioned in the Weaknesses, please pay attention to following questions:

1. How were the 136 websites selected for Online-Mind2Web? Were criteria like regional diversity, traffic volume, or technical complexity (e.g., SPAs, JavaScript-heavy sites) explicitly considered? Could you provide statistics on website categories (e.g., e-commerce, healthcare, social media) to demonstrate representativeness?

2. The step-count-based difficulty categorization is intuitive, but how did annotators resolve discrepancies in defining a "step"? For example, does a single API call count as one step or multiple? Were inter-annotator agreement metrics reported?

3. In WebJudge’s key screenshot identification, why was δ set to 3? Could you share results from sensitivity tests (e.g., δ=2 or 4) and their impact on agreement with human judgment?

4. The decision to prohibit Google Search during agent evaluation is understandable for isolating navigation skills, but many real-world tasks rely on search. How would including search affect agent performance metrics, and did you consider a hybrid evaluation (with/without search)?

**Reasons To Accept:**

1. The paper provides a much-needed reality check for the field by exposing shortcomings in popular benchmarks like WebVoyager, such as task simplicity and evaluation bias. The introduction of Online-Mind2Web, with its emphasis on diverse, real-world tasks and manual evaluation, addresses critical gaps in existing benchmarks. The task categorization by difficulty (easy/medium/hard) adds nuance to performance analysis.

2. WebJudge’s three-step evaluation process (key point identification, key screenshot selection, outcome judgment) is a robust innovation. Achieving 85% agreement with human judgment significantly improves evaluation scalability compared to manual methods, while mitigating token overload and intermediate-step neglect.

3. The comparative study of five agents (Operator, SeeAct, etc.) on the new benchmark offers valuable insights. For example, Operator’s 61% success rate contrasts sharply with prior claims on WebVoyager, grounding the discussion in realistic performance metrics. The error analysis (e.g., Operator’s struggles with numerical constraints) provides actionable directions for improvement.

4. The dataset construction process, which filters out invalid tasks and incorporates live website evaluation, ensures high ecological validity. The manual validation of 650 initial tasks and inclusion of 75 new tasks demonstrate meticulous attention to benchmark quality.

**Reasons To Reject:**

Weaknesses:
1. While Online-Mind2Web is praised for diversity, the paper does not detail how website selection was prioritized (e.g., traffic metrics, domain representation). Are the 136 websites representative of global web complexity (e.g., regional variations, niche platforms)?

2. The "no Google Search" constraint may not reflect real-world agent usage, where search is often a critical tool. Clarifying whether this constraint is experimental or prescriptive is necessary.

3. The reliance on single GPT-4o for WebJudge introduces bias risks, as LLM evaluators may inherit model-specific limitations. Exploring multiple LLM judges (e.g., Qwen3, Claude, DeepSeek-R1) to collaborate would enhance accessibility and reduce dependency on proprietary models.

---

> ### Author Response · Authors · 2025-06-01
> **Author Response to Reviewer inKQ (Part 1/2)**
>
> We thank the reviewer inKQ for the constructive comments. We are glad you find Online-Mind2Web is diverse, grounded in real-world tasks, and recognize the effectiveness of WebJudge. We will address your questions in the following paragraphs.
>
> ---
>
> **W1 and Q1: The selection of websites.**
>
> For website selection, we followed a similar approach to Mind2Web, prioritizing selecting popular websites based on traffic volume reported by SimilarWeb(https://www.similarweb.com/). Specifically, for tasks from the Mind2Web and Mind2Web-Live datasets, we chose tasks from different websites as much as possible. (Notably, tasks from the original Mind2Web and Mind2Web-Live datasets were created based on website traffic volume in the U.S., as ranked by SimilarWeb. Please see more details in the paper [1].) For the newly constructed tasks by us, we similarly selected previously unpresented but popular websites, ranked by traffic volume, according to SimilarWeb. To further ensure diversity, we also included 14 tasks from niche websites. Based on this construction process, we ultimately obtained 300 tasks covering 136 unique websites. You can see the task popularity (traffic volume) distribution in Figure A.1 in the appendix. We will clarify the selection process of websites in our revision.
> Our primary selection criterion was traffic volume, as it most closely reflects real-world scenarios where users typically perform tasks on popular websites. At this stage, we didn't consider technical complexity factors. However, we still included websites from different regions, such as those in the UK, as well as a small number of niche websites in our task set to enhance diversity and representation.
>
> Here are the statistics of the website category. We categorize them into 12 domains
> including Shopping & E-Commerce, Food & Recipes, Housing & Real Estate, Finance & Investment, Health & Medical, Travel & Transportation, Entertainment & Media, Technology, Education, Government & Services, Jobs & Careers and Other. The table below shows the number of websites for each domain.
>
> | Domain | Number of Websites |
> |--------------------------|--------------------|
> | Shopping & E-Commerce | 24 |
> | Food & Recipes | 3 |
> | Housing & Real Estate | 8 |
> | Finance & Investment | 8 |
> | Health & Medical | 9 |
> | Travel & Transportation | 16 |
> | Entertainment & Media | 18 |
> | Technology | 12 |
> | Education | 15 |
> | Government & Services | 11 |
> | Jobs & Careers | 4 |
> | Other | 8 |
>
> [1] Deng, Xiang, et al. "Mind2web: Towards a generalist agent for the web." Advances in Neural Information Processing Systems 36 (2023): 28091-28114.
>
> ---
>
> **W2 and Q4: The "no Google Search" constraint.**
>
> We thank the reviewer for pointing out this important discussion!  Please see our general response.
>
> ---
>
> **W3: The bias risks and dependency on proprietary models.**
>
> We understand the concern regarding bias and reliance on proprietary models. To address this, we have synthesized training data from multiple models and trained a lightweight WebJudge-7B, based on Qwen2.5-VL-7B. This model not only achieves a high agreement rate with human judgments (87%) but also shows a small success rate gap of just 3.9% compared to human evaluation. Furthermore, it generalizes well to out-of-distribution (OOD) benchmarks, even outperforming rule-based methods. Please see our general response for more details. We will add the details in our final manuscript.

---

> > ### Author Response · Authors · 2025-06-01
> > **Author Response to Reviewer inKQ (Part 2/2)**
> >
> > **Q2: Step-count-based difficulty categorization and “does a single API call count as one step or multiple?”.**
> >
> > The step count is primarily calculated by summing the number of Click and Type actions. For newly created tasks or tasks that we adapted from Mind2Web and Mind2Web-Live, the required number of steps is provided by the task annotator, who is asked to specify the minimum number of steps needed to complete the task in order to better evaluate agent efficiency (i.e., whether the agent performs unnecessary actions). For tasks sourced from Mind2Web and Mind2Web-Live, we retain the original step counts. When calculating the number of steps for agents, we count it as the number of actions (i.e., clicks on elements or coordinates, and typing). For different agents, a single action may involve varying numbers of API calls. For example, SeeAct requires planning first and then selecting an interactive element through several rounds of a multiple-choice format, so in general, one action often results in multiple API calls.
> >
> > ---
> >
> > **Q3: The impact of threshold δ.**
> >
> > Thanks for this interesting question! Different thresholds δ (i.e., screenshots with a relevance score of task completion above or equal to a threshold δ are filtered as key screenshots) will result in different inputs for the final outcome judgement. Specifically, a larger δ will significantly reduce the number of screenshots but also lose a lot of information for judging. In contrast, smaller δ will maintain most of the intermediate screenshots but also increase the issue of token overhead and suppress the model's context. We conducted experiments on Operator’s trajectories (300 tasks) to see the influence of threshold δ. As shown in the following table, we observe that δ=3 achieves the highest agreement with human judgments while  δ=2 and δ=4 show a noticeable drop in agreement. δ=2 introduces too many screenshots for the final outcome judgment, overwhelming the model and making it difficult to focus on the most informative ones. In contrast, δ=4 ignores many important intermediate screenshots, leading to the loss of critical information. For all experiments in the paper, we used the threshold δ=3 throughout. We did not set different thresholds for different agents in order to ensure consistency and general applicability.
> >
> > | WebJudge | δ=2 | δ=3 | δ=4 |
> > |-------------|-------|-------|-------|
> > | Agreement | 79.3% | **81.8%** | 77.0% |

---

> > > ### Author Response · Authors · 2025-06-08
> > > **Gentle Reminder from Authors**
> > >
> > > Dear Reviewer inKQ,
> > >
> > > As the end of discussion period is approaching, we would like to gently remind you of our responses to your comments. We wonder whether your concerns have been addressed and appreciate any further questions or comments you might have.
> > >
> > > Sincerely,
> > >
> > > Authors of Submission 1009

---

### Author Response · Authors · 2025-06-01
**General Response to Reviewers (Part 1/2)**

We appreciate all reviewers for their time and efforts! We are glad that reviewers find our benchmark valuable and recognize the effectiveness of WebJudge. We address some common questions raised by reviewers below:

**Q1: Could WebJudge be built upon an open-source model?**

We understand that relying on closed-source models can introduce several issues, such as bias and limited accessibility (e.g., high cost and latency) mentioned by reviewer inKQ, as well as the issue of inconsistency over time raised by reviewer 8d2e. To relieve these issues and further improve reliability, we train a lightweight key screenshot identification model called WebJudge-7B. Specifically, we synthesize training data from the key screenshot identification stages of several models, including GPT-4o, GPT-4.1-mini, Claude 3.7, and Qwen2.5-VL-72B. We use trajectories from SeeAct, Browser Use, and Claude Computer Use 3.5 for training, and reserve Agent-E, Claude Computer Use 3.7, and Operator as held-out evaluation sets. During inference, the trained model summarizes each screenshot and assigns a relevance score, while key point identification and outcome judgment are still handled by an LLM (e.g., GPT-4o). This reduces the number of API calls in evaluating each trajectory, from being proportional to the trajectory length to a fixed number of two calls per trajectory, and also mitigates the bias problem and the randomness compared with using one single model for the key screenshot identification stage.

Since the cost of o4-mini is significantly lower than GPT-4o (half that of GPT-4o) and we observed better performance in key point identification and outcome judgment stages when using o4-mini. Therefore, we use o4-mini for these stages of WebJudge-7B. For reference and to ensure fair comparison, we also report the results of WebJudge (o4-mini).

| **Agent** | **Human Eval** | **WebJudge (GPT-4o)** | **WebJudge (GPT-4o)** | **WebJudge (o4-mini)** | **WebJudge (o4-mini)**| **WebJudge-7B** | **WebJudge-7B** |
|---------------------------|----------------|------------------------|------|------------------------|------|------------------|------|
| | | **SR** | **AR** | **SR** | **AR** | **SR** | **AR** |
| SeeAct | 30.7 | 39.8 | 86.7 | 30.0 | 85.3 | 28.0 | 86.0 |
| Agent-E | 28.0 | 34.7 | 86.0 | 27.0 | 86.3 | 26.0 | 87.3 |
| Browser Use | 30.0 | 40.1 | 81.4 | 26.0 | 89.3 | 24.3 | 88.3 |
| Claude Computer Use 3.5 | 29.0 | 35.8 | 86.3 | 24.0 | 87.0 | 26.0 | 89.7 |
| Claude Computer Use 3.7 | 56.3 | 59.7 | 79.1 | 47.3 | 82.3 | 48.7 | 84.3 |
| OpenAI Operator | **61.3** | **71.8** | 81.8 | **58.3** | 83.7 | **59.0** | 86.3 |
| **Avg Gap/AR** | | 7.8 | 83.6 | 3.8 | 85.7 | 3.9 | 87.0 |

The SR denotes success rate and AR indicates agreement rate.
We can observe that WebJudge, powered by a stronger model like o4-mini, can further unleash its potential, achieving approximately 86% agreement with human judgment and maintaining only a 3.8% gap from actual success rates. We believe this result is also valuable for the community to know, as it demonstrates that WebJudge’s prediction of success is closely aligned with human evaluation on Online-Mind2Web. Moreover, compared with WebJudge (o4-mini), WebJudge-7B achieves a higher agreement rate of 87%, with a small performance gap of only 3.9%. These results indicate that WebJudge-7B is more reliable and accessible for the community to use.

---

> ### Author Response · Authors · 2025-06-01
> **Q1 Part 2: The generalization ability of WebJudge-7B**
>
> Considering that WebJudge-7B may overfit on our benchmark and thus has limited applicability, we further test it on AgentRewardBench [1], which comprises 1302 trajectories spanning 5 popular benchmarks: WebArena (WA), VisualWebArena (VWA), AssistantBench (AB), WorkArena (Work), and WorkArena++ (Wk++).
>
> GPT-4o (Ally Tree) represents the prior SOTA results in the paper[1], which is based on the accessibility tree. Rule-based denotes the official rule-based evaluation method.  “*” indicates results are taken from the paper [1]. We follow previous settings and report the same metric of precision on AgentRewardBench [1], as shown in the following table.
>
> | **Methods** | **AB** | **VWA** | **WA** | **Work** | **Wk++** | **Overall** |
> |-----------------------|--------|---------|--------|----------|----------|-------------|
> | *Rule-based* | 25.0 | **85.2**| 79.0 | 100.0 | 83.3 | 83.8 |
> | Autonomous Eval* | 83.3 | 61.2 | 67.6 | 96.4 | 59.3 | 67.6 |
> | GPT-4o (A11y Tree)* | 77.8 | 63.0 | 70.2 | 94.6 | 63.0 | 69.8 |
> | WebJudge (GPT-4o) | 66.7 | 69.8 | 72.6 | 92.3 | 75.0 | 73.7 |
> | WebJudge-7B | 80.0 | 66.7 | 77.5 | 100.0 | 70.0 | 75.7 |
> | WebJudge (o4-mini) | **100.0**| 74.5 | **81.2**| **100.0**| **90.0** | **82.0** |
>
> The table below illustrates the success rate gaps of various methods compared to human evaluation across several benchmarks.
>
> | Agent | Human Eval | Human Eval | Human Eval | GPT-4o (A11y Tree) | GPT-4o (A11y Tree) | GPT-4o (A11y Tree) | Rule-based | Rule-based | Rule-based | WebJudge (GPT-4o) | WebJudge (GPT-4o) | WebJudge (GPT-4o) | WebJudge (o4-mini) | WebJudge (o4-mini) | WebJudge (o4-mini) | WebJudge-7B | WebJudge-7B | WebJudge-7B |
> |---------------|----------------|----------------|----------------|-------------------|-------------------|-------------------|---------------|----------------|----------------|--------------------|-------------------|-------------------|--------------------|--------------------|-------------------|-------------|-------------|-------------|
> | | VWA | WA | Wk++ | VWA | WA | Wk++ | VWA | WA | Wk++ | VWA | WA | Wk++ | VWA | WA | Wk++ | VWA | WA | Wk++ |
> | Claude 3.7 | 28.3 | 55.1 | 18.4 | 34.8 | 64.1 | 20.7 | 23.9 | 30.8 | 8.1 | 31.5 | 52.6 | 16.1 | 18.5 | 33.3 | 3.5 | 22.8 | 44.9 | 10.3 |
> | GPT-4o | 35.9 | 42.3 | 18.4 | 47.8 | 50.0 | 11.5 | 17.4 | 25.6 | 4.6 | 31.5 | 46.2 | 9.2 | 20.7 | 32.1 | 3.5 | 29.4 | 38.5 | 4.6 |
> | Llama 3.3 | -- | 22.4 | 9.2 | -- | 27.6 | 5.8 | -- | 18.4 | 3.5 | -- | 30.3 | 3.5 | -- | 14.5 | 1.2 | -- | 23.7 | 1.2 |
> | Qwen2.5-VL | 21.7 | 33.3 | 13.8 | 34.8 | 52.6 | 14.9 | 17.4 | 29.5 | 11.5 | 30.4 | 44.9 | 8.1 | 20.7 | 29.5 | 3.5 | 22.8 | 35.9 | 6.9 |
> | *Gap* | -- | -- | -- | 10.5 | 10.3 | 3.4 | 9.1 | 12.2 | 8.0 | 5.4 | 6.5 | 5.7 | 8.7 | 10.9 | 12.0 | 4.4 | 4.5 | 9.2 |
> | *Avg Gap* | -- | -- | -- | | 8.1 | | | 9.8 | | | **5.9** | | | 10.5 | | | 6.0 | |
>
> We can find that WebJudge-7B achieves both high precision (75.7%) and a smaller success rate gap (6.0%), making it a reliable and scalable evaluator for further research. Overall, these results further emphasize the effectiveness and reliability of WebJudge-7B as an automatic evaluator, making it a valuable tool for rapid iteration in agent development and evaluation.
>
> [1] Lù, Xing Han, et al. "AgentRewardBench: Evaluating Automatic Evaluations of Web Agent Trajectories." arXiv preprint arXiv:2504.08942 (2025).

---

> > ### Author Response · Authors · 2025-06-01
> > **General Response to Reviewers (Part 2/2)**
> >
> > **Q2: The constraints of Google Search.**
> >
> > There are two main reasons why we restrict agents from using Google Search.
> > - First, we aim to evaluate the agent's navigation capabilities as much as possible while minimizing the influence of other factors, such as shortcutting via Google search.
> > - Second, and more importantly, we observe that without this restriction, agents tend to use Google Search to navigate to other alternative websites when they encounter difficulties on the designated one, resulting in task completion on entirely different websites. This will lead to unfair comparisons among agents, as different websites have varying designs and functions, which may impact the difficulty of the task. To eliminate such confounding factors and enable a fair, apples-to-apples comparison, we prohibit the use of Google Search. However, we acknowledge that using search is a natural strategy for solving web tasks. To better understand its impact, we evaluate the Browser-Use agent without this restriction (this agent is most likely to take shortcuts via search). Interestingly, even when it is allowed to use Google Search, its performance did not improve by a lot (26% vs 31%), suggesting that current agents still struggle to handle real user tasks on real-world websites. The small improvement also highlights that Online-Mind2Web tasks are less likely to be solved by shortcuts, unlike WebVoyager's tasks, 50% of which can be completed using such shortcuts (See our detailed discussion in Section 3.1).
> >
> > | Browser-Use (w/o Google Search) | Browser-Use (w/ Google Search) |
> > |---------------------------------|--------------------------------|
> > | 26% | 31% |
> >
> > ---
> >
> > **Q3: The maintenance strategy of Online-Mind2Web.**
> >
> > We acknowledge that maintaining a benchmark is both challenging and time-consuming, but it is also critically important. As websites evolve over time, features may be added or removed and webpage structures may also change, potentially rendering certain tasks outdated or unsolvable. In fact, this is a common issue for all online web benchmarks. The dynamic nature of the web is a double-edged sword: while it brings benchmarks closer to real-world scenarios, it also significantly increases the difficulty of maintenance.
> > To address this, we plan to adopt a strategy that involves close collaboration with the community. Specifically, we will actively gather feedback from the web agent community because they are familiar with the tasks through conducting evaluations and investigating failure cases. Additionally, whenever a widely recognized and powerful agent is released, we will conduct human evaluations on its failure cases to ensure that these tasks remain solvable. This will not only help us verify the validity of the tasks but also ensure accurate evaluation of the agent. In the future, we will consider a human-in-the-loop approach that can automate the maintenance process as much as possible, while still involving humans to ensure accuracy and quality.

---

### Decision · Program_Chairs · 2025-07-08

**Decision:**

Accept

**Comment:**

This paper provides an insightful evaluation of recent web navigation agents, effectively highlighting their capabilities and limitations. The proposed WebJudge framework, leveraging an LLM as an evaluator, is methodologically sound and well-conceived. The comparative analysis of recent web agents adds valuable depth to the paper. However, the study would benefit significantly from a more thorough and rigorous meta-evaluation of the LLM-based judge itself, providing clearer validation and reinforcing the reliability of the evaluation framework.